# Counterfactual Evolution of Multimodal Datasets via Visual Programming

**Minghe Gao**[1,2][*] **Zhongqi Yue**[3][*] **Wenjie Yan**[1]**, Yihao Hu**[4]**, Wei Ji**[5]
**Siliang Tang**[1]**, Jun Xiao**[1]**, Tat-Seng Chua**[2]**, Yueting Zhuang**[1]**, Juncheng Li**[1][†]

[1]Zhejiang University     [2]National University of Singapore
[3]Nanyang Technological University     [4]Hainan University     [5]Nanjing University
{minghegao,22551068,siliang,junx,yzhuang,junchengli}@zju.edu.cn
nickyuezhongqi@gmail.com, 20223003513@hainanu.edu.cn
weiji0523@gmail.com, chuats@comp.nus.edu.sg

## Abstract

The rapid development of Multimodal Large Language Models (MLLMs) poses increasing demands on the diversity and complexity of multimodal datasets. Yet manual annotation pipelines can no longer keep pace. Existing augmentation methods often follow fixed rules and lack verifiable control over sample diversity and reasoning complexity. To address this, we introduce Scalable COunterfactual Program Evolution (**SCOPE**), a framework that uses symbolic Visual Programming to guide program evolution via counterfactual reasoning. **SCOPE** performs the three steps of counterfactual inference: (1) Abduction, by generating verifiable programs to model reasoning associations; (2) Action, by intervening on program structure along three axes—reasoning path, visual context, and cross-instance composition; and (3) Prediction, by categorizing evolved instances by difficulty, structure, and input multiplicity. Based on this process, we build *SCOPE-Train* and *SCOPE-Test*, evolving benchmarks with expert validation. To support training, we propose MAP, a curriculum learning strategy that aligns model capacity with sample difficulty. Experiments show that **SCOPE** improves reasoning performance, exposes model blind spots, and enhances visual dialog capabilities.

## 1   Introduction

Multimodal datasets [37, 11, 21, 5, 28] have been foundational to the development of powerful MLLMs. Recent works [10, 22, 35, 36] have devoted substantial effort to collecting and curating data for both the pretraining and evaluation, aiming to enhance the quality of raw datasets. An ideal dataset should be sufficiently diverse and scalable, with controllable attributes such as difficulty, input modality, and the ability to explicitly reflect reasoning processes. However, existing datasets are mostly static, lack diversity, or rely heavily on manual annotation that can no longer keep pace with model development. The growing mismatch between the speed of dataset evolution and model advancement has become a critical bottleneck in developing more generalizable and robust MLLMs. A systematic dataset evolution framework is urgently needed to address the limitations posed by low-quality data resources.

Several recent efforts have attempted to modify existing datasets, which can be broadly categorized into three approaches. **A:** Prompt-based transformations using large models, such as MAmmoTH-VL [16], CoSyn [41], and OmniSearch [20]. These methods feed original VQA or image-question

---

[*]Equal contribution.
[†]Corresponding author.

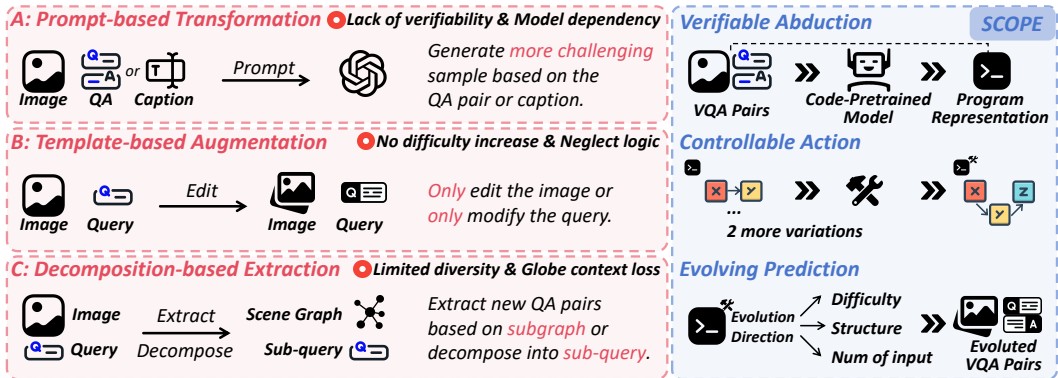

Figure 1: Comparison between current data evolution methods and **SCOPE**.

pairs into a large model and prompt it to generate more challenging examples. However, this approach lacks verifiability, heavily depends on the model's capabilities, and contradicts the original intention of improving models through data rather than relying on the model to generate data. **B:** Template-based augmentation, as seen in VLB [42] and SK-VQA [38], where predefined operations are applied to edit questions, modify images, or incorporate external knowledge to produce more diverse samples. Nonetheless, these methods do not increase sample difficulty and often overlook the underlying reasoning logic of the original problems, potentially rendering the augmented data ineffective. **C:** Decomposition-based local extraction, exemplified by AGQA-Decomp [13] and Provision [45]. These methods split complex questions into sub-questions or represent images as scene graphs to generate fine-grained samples from a local perspective. Yet, this leads to the loss of global context, accumulation of local biases through multiple iterations, and limited diversity. We summarize these limitations in Figure 1. Overall, they remain far from the ideal of dataset evolution—namely, verifiable transformations, controllable difficulty enhancement, and diverse sample generation.

To address these limitations, we introduce a counterfactual approach and adopt Visual Programming [39, 17] as the core tool. Visual Programming leverages code-pretrained models to synthesize executable programs for vision tasks, where vision-language reasoning is represented through symbolic code. This allows us to explicitly construct and trace a reasoning chain in a fully interpretable and verifiable manner throughout the entire process, laying the foundation for structured counterfactual inference through abduction, action, and prediction.

Specifically, we propose **SCOPE**, a three-stage dataset evolution framework: **In the verifiable abduction stage**, we generate Python programs for each vision-language task and link the verifiable code statements explicitly to the underlying reasoning paths. **In the controllable action stage**, we enable targeted extensions along three dimensions: reasoning path expansion, visual context editing, and cross-instance composition, thereby allowing for systematic and proactive intervention in the reasoning process. **In the evolving prediction stage**, instances are categorized based on reasoning difficulty, structural variation, and input multiplicity relative to the original samples, enabling continuous and diverse evolution from the source data. **Finally**, we propose MAP, a curriculum strategy that enables fine-grained scheduling, targeted module activation, and adaptive difficulty weighting, forming a training paradigm coupled with **SCOPE**'s program-centric design.

Building on **SCOPE**, we construct the benchmark sets *SCOPE-Train* and *SCOPE-Test*. Unlike static benchmarks, they support continuous evolution toward increased diversity and complexity. The test set was curated and verified by domain experts, achieving high approval rates during manual review. Static and comparative results against prior benchmarks are shown in Figure 2a and 2b.

Extensive experiments validate the effectiveness of **SCOPE** across multiple dimensions: (1) *SCOPE-Test* reveals systematic blind spots in current models that remain undetected under conventional benchmarks, with expert review confirming the high quality of generated samples. (2) Models of varying scales benefit from **SCOPE**, exhibiting a clear scaling trend in reasoning performance. (3) Models trained with **SCOPE** demonstrate strong easy-to-hard generalization and achieve notable gains on visual dialog tasks. (4) The MAP curriculum learning framework further enhances training efficiency by aligning sample difficulty with model capacity across iterative expansion rounds. In summary, our contributions are threefold:

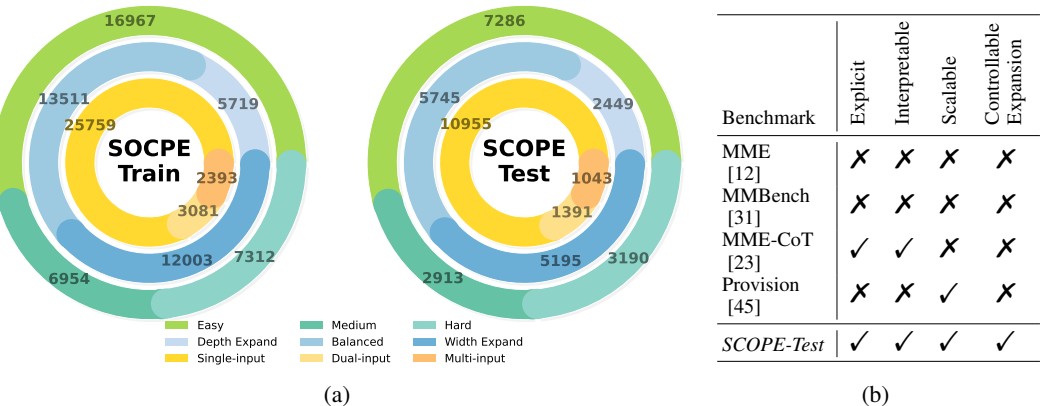

| Benchmark | Explicit | Interpretable | Scalable | Controllable Expansion |
|---|:---:|:---:|:---:|:---:|
| MME [12] | ✗ | ✗ | ✗ | ✗ |
| MMBench [31] | ✗ | ✗ | ✗ | ✗ |
| MME-CoT [23] | ✓ | ✓ | ✗ | ✗ |
| Provision [45] | ✗ | ✗ | ✓ | ✗ |
| *SCOPE-Test* | ✓ | ✓ | ✓ | ✓ |

(a)         (b)

Figure 2: (a) Data distribution of *SCOPE-Train* and *SCOPE-Test*. (b) Benchmark comparison.

- We propose **SCOPE**, which constructs verifiable associations, enables controllable interventions, and supports counterfactual evolution through a symbolic program.
- We construct *SCOPE-Train* and *SCOPE-Test*, a continuously evolving benchmark supporting scalable training and fine-grained diagnostic evaluation.
- We introduce the Memory and Attention Path (MAP) Learning, which aligns training with **SCOPE**'s structured difficulty progression, improves generalization on easy-to-hard samples.

## 2 Related Work

### 2.1 Visual Programming Edition.

Visual Programming [39, 17] has recently emerged as a promising paradigm for generating interpretable and modular reasoning pipelines in multimodal tasks by utilizing neural symbols or Python modules for task synthesis and execution. Unlike black-box end-to-end models or purely textual Chain-of-Thought (CoT) generation, it enables fine-grained control over the reasoning process through code editing. Prior work has explored its potential from various angles: De-fine [15] integrates feedback loops to improve code generation quality, while VPD [19] and Fact [14] transform execution traces into explicit reasoning paths for distillation learning. Although Provision [45] also uses Python code for data generation, it remains restricted by predefined scene graphs and lacks broader data-space adaptability. In contrast, our **SCOPE** framework advances Visual Programming from a generation tool to a controllable benchmark construction mechanism. By supporting both program-level augmentation and iterative evolution, **SCOPE** enables structured dataset expansion.

### 2.2 Counterfactual Data Generation

Counterfactual data generation [33, 4] has emerged as a promising strategy for improving model robustness and mitigating language biases in multimodal learning. Approaches such as COCO-Counterfactuals [25] and Counterfactual Prompt Learning (CPL) [18] generate hypothetical variants of image-text pairs or prompts to challenge spurious correlations. Notably, Counterfactual Samples Synthesizing (CSS) [6] introduces a model-agnostic training scheme that creates counterfactual examples by masking critical objects in images or keywords in questions, enhancing both visual-explainable and question-sensitive capabilities of VQA models. However, most existing methods rely on heuristic rules and lack fine-grained control over reasoning complexity. In contrast, our **SCOPE** framework integrates counterfactual generation into a structured causal learning paradigm via symbolic Visual Programming, enabling interpretable and scalable dataset evolution.

## 3 Method

To enable controllable dataset evolution, we introduce **SCOPE**, as illustrated in Figure 3. Starting from an image–question–answer triplet, we employ Visual Programming to generate a Python program, retaining only those that execute correctly (Section 3.1). This executable representation serves as the foundation for structured expansion: reasoning path expansion, visual context editing, and

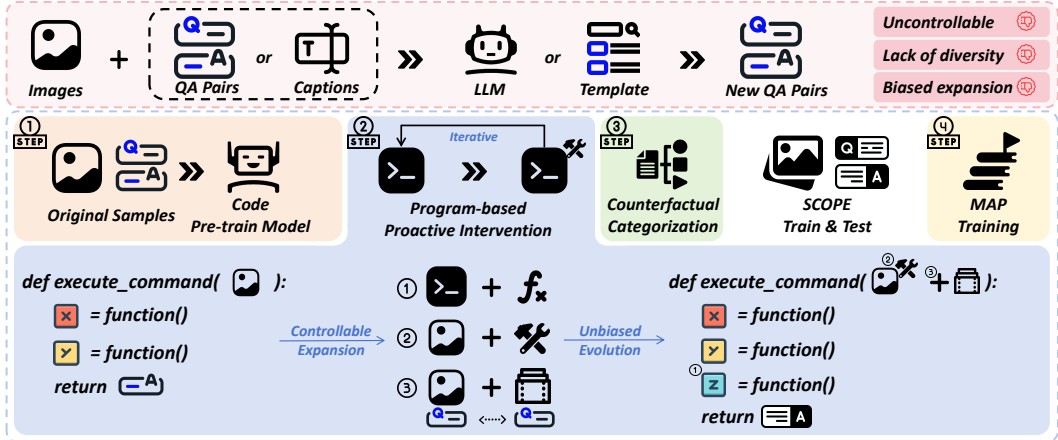

Figure 3: The pipeline of **SCOPE**: we generate executable visual programs for the given instances, enabling iterative controllable expansion and categorization of their evolutionary progression.

cross-instance composition (Section 3.2). The expanded instances are then categorized by reasoning difficulty, structural variation, and input multiplicity (Section 3.3). Finally, we introduce MAP that aligns model curriculum training with the structured evolution of the dataset (Section 3.4).

## 3.1 Verifiable Abduction through Visual Programming Generation

Programs, with their explicit logic and variable dependencies, offer greater verifiability and interpretability than natural language. Leveraging Visual Programming, we translate image–question pairs into executable Python code, enabling fine-grained manipulation of inputs and functions. This symbolic form serves as a verifiable abduction of the reasoning path, ensuring structural coherence and semantic grounding. Moreover, code-based reasoning aligns with the strengths of pretrained models in program synthesis, supporting efficient knowledge integration and streamlined expansion.

More formally, given an image $x$ and its corresponding query $q$, we use a program generator $\pi$ (e.g., GPT-4o) to produce executable code $z = \pi(x, q)$. This program is then evaluated by an execution engine $\phi(z, x)$ that applies $z$ to the input. If the program output matches the ground-truth $a$, the instance is retained. Each valid instance is represented as a four-tuple $(q, x, z, a)$—comprising the question, image, program, and answer—which serves as the foundation for all subsequent expansions. Additional implementation details, including API calls and prompts, are provided in the **Appendix**.

## 3.2 Controllable Action by Proactive Intervention

To address the limitations of heuristic or template-based augmentation. **SCOPE** introduces a structured expansion paradigm grounded in symbolic reasoning. Through program-level control, it supports a systematic intervention from simple to complex instances via logic composition, visual variation, and cross-source integration. Specifically, we propose three expansion strategies:

**Reasoning Path Expansion.** We use a modular API library of atomic functions (e.g., `find`, `exist`) that encapsulate distinct reasoning steps. These functions can be composed or nested within program structures to increase reasoning depth or width. For example, extending "How many apples are there?" to "How many red apples are there?" involves inserting a `verify_property` call. This compositional scheme ensures extensibility and preserves logical traceability for analysis.

**Visual Context Editing.** To diversify visual contexts, we perform localized edits on segmented image entities using tools like SAM [24]. By replacing, altering, or removing objects, we generate fine-grained visual variations while holding the program structure constant. This method supports incremental complexity increases in visual reasoning and mitigates biases arising from overexposure to image, promoting controllable generalization.

**Cross-Instance Composition.** To enhance cross-domain diversity, **SCOPE** expands single-instance tasks into multi-image or multi-source settings by combining examples from different datasets with semantically aligned questions. This approach maximizes code reuse through shared program

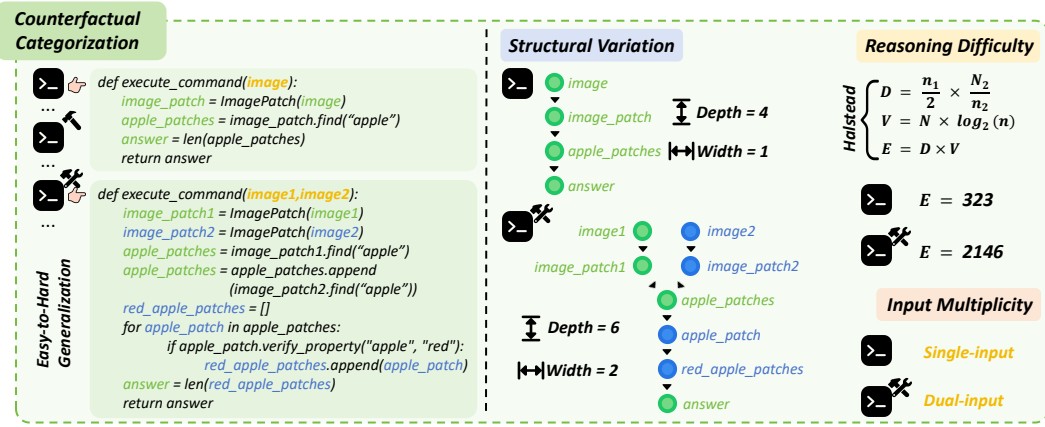

Figure 4: **SCOPE** categorizes the data based on changes in task difficulty, program structure, and input modalities before and after code expansion, involved images can be found in **Appendix**.

structures and reduces visual redundancy. It enables composition across heterogeneous domains, facilitating richer task representations while avoiding contamination typical in monolithic datasets.

### 3.3 Evolving Prediction through Counterfactual Categorization

Following the intervention, each instance is accompanied by a structured program representation. Unlike natural language, code offers explicit syntactic and semantic regularities, enabling systematic comparison with the original program and categorization: reasoning difficulty, dependency structure, and input multiplicity. The figure 4 illustrates the analysis of these three dimensions.

**Reasoning Difficulty.** The complexity of a program reflects the cognitive effort required to solve the corresponding task. We adopt the Halstead effort from software engineering to assess difficulty:

$$E = D \times V \quad \text{where} \quad V = N \log n, \quad n = n_1 + n_2, \quad D = \left(\frac{n_1}{2}\right) \times \left(\frac{N_2}{n_2}\right)$$

Here, $n_1, n_2$ are the counts of unique operators and operands, and $N_1, N_2$ are their total occurrences, with $n = n_1 + n_2$, $N = N_1 + N_2$. A higher $E$ reflects increased reasoning complexity. We use this metric in different threshold to classify the instance into three levels: easy, medium, and hard.

**Structural Variation.** We represent each program as a directed acyclic graph (DAG), where nodes denote variables and edges capture dependency relations. Depth (longest path) and width (maximum fan-in) characterize the structure. We define depth and width expansion factors as relative increases in these values post-expansion. An expansion is classified as depth, width, or balanced, depending on which factor dominates or whether both expand comparably. This taxonomy ensures a balanced representation of program complexities, avoiding overrepresentation of singular structural patterns.

**Input Multiplicity.** Expansions may also vary the input modality, transitioning from single-image to multi-image VQA tasks. We track the number of distinct input sources per instance as a proxy for input complexity. This metric enables separation of reasoning complexity from input diversity and supports nuanced analysis of how cross-modal integration contributes to generalization.

### 3.4 Memory and Attention Path Curriculum Learning

**SCOPE** offers a natural foundation for curriculum learning by supporting the generation of samples with progressively increasing difficulty. To fully leverage this property, we design a progressive curriculum learning framework that aligns with **SCOPE**'s controllable evolution paradigm.

We first define a curriculum scheduler that ranks training samples by increasing difficulty, prioritizing (1) input multiplicity, (2) structural variation, and (3) reasoning complexity. For example, single-image, depth-oriented, low-complexity samples are treated as easiest, while multi-image, width-oriented, high-effort samples are considered hardest. In structural variation, we heuristically regard width expansions as more challenging than depth ones, a rationale that will be further analyzed in the

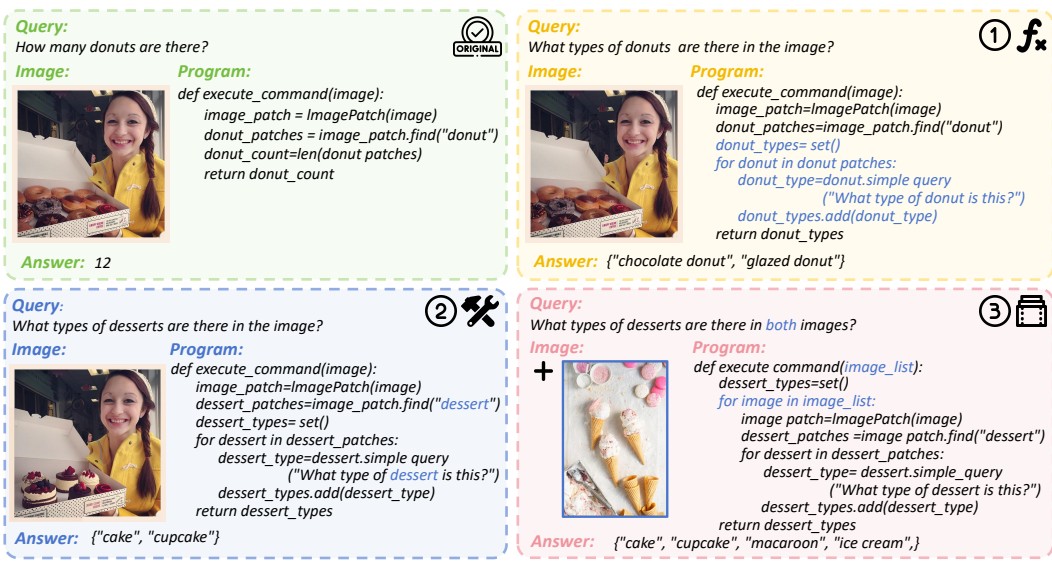

Figure 5: An example of **SCOPE** evolution and the annotation format is provided in the **Appendix**.

experimental section. Next, we introduce two adaptive modules: a memory-augmented branch $\mathcal{M}$ for depth-oriented tasks, and a parallel attention branch $\mathcal{A}$ for width-oriented tasks. A gating function $\mathcal{F}$ selects the appropriate path based on the sample type.

$$\mathcal{F}(x_i) = \begin{cases} \mathcal{M}(x_i), & \text{if } r_i = depth, \\ \mathcal{A}(x_i), & \text{if } r_i = width, \\ \frac{1}{2}\mathcal{M}(x_i) + \frac{1}{2}\mathcal{A}(x_i), & \text{if } r_i = balanced. \end{cases}$$

Third, we incorporate a difficulty-aware loss function. In addition to the base training objective, we apply a normalized Halstead effort score (from Section 3.3) as a dynamic weight to modulate each sample's contribution to the loss.

$$\mathcal{L}_i = \tilde{E}_i \cdot \mathcal{L}_{\text{base}}(x_i), \quad \tilde{E}_i = \frac{E_i}{\sum_{j=1}^n E_j}$$

$$\mathcal{L}_{\text{total}} = \frac{1}{n}\sum_{i=1}^n \tilde{E}_i \cdot L_{\text{base}}(x_i)$$

This encourages the model to adaptively allocate learning focus according to reasoning complexity. Together, these components enable effective curriculum-style training over **SCOPE**'s structured expansions, yielding models with improved reasoning capabilities.

## 4 **SCOPE** Benchmark

**Dataset collection and construction.** The benchmark is built by applying the **SCOPE** framework to perform program-guided expansions over samples from six widely used vision-language datasets: SEED-Bench2 [26], MME [23], MM-Bench [31], GQA [21], OK-VQA [34], and Tally-QA [2]. To avoid information leakage and ensure fair evaluation, only the official *dev or test* splits are used for expansion. Detailed statistics on dataset composition are provided in the **Appendix**, and a static distribution of different attributes is shown in Figure 2a. Based on Halstead effort scores, we define reasoning difficulty using two empirical thresholds: 4000 and 6000, which separate instances into *easy*, *medium*, and *hard* levels. The final dataset is divided into *SCOPE-Train* and *SCOPE-Test* with a 70:30 ratio. Compared with traditional static benchmarks, **SCOPE** enables program-centric, multi-round augmentation and difficulty-aware diagnostics. A comparative summary with prior datasets is presented in Figure 2b. We also provide a statistical overview of the question distribution across the three evolution stages of SCOPE, which can be found in Table 1.

Table 1: Question distribution statistics of **SCOPE** across training and test sets.

| Question Type | *SCOPE-Train* (Number / Proportion (%)) | *SCOPE-Test* (Number / Proportion (%)) |
|---|---|---|
| Counting | 4,309 / 13.8% | 1,579 / 11.8% |
| Object Attribute (color, texture, shape) | 6,471 / 20.7% | 2,534 / 18.9% |
| Spatial Relationships | 5,377 / 17.2% | 2,589 / 19.3% |
| OCR-based Reasoning | 6,127 / 19.6% | 2,411 / 18.0% |
| Multi-hop | 5,903 / 18.9% | 2,958 / 22.1% |
| Others | 3,046 / 9.8% | 1,318 / 9.9% |
| **Total** | **31,233 / 100.0%** | **13,389 / 100.0%** |

Table 2: Comparison of various models on *SCOPE-Test*.

| Model | Reasoning Difficulty | | | Structural Variation | | | Input Multiplicity | | Overall |
|---|---|---|---|---|---|---|---|---|---|
| | Easy | Medium | Hard | Depth | Balanced | Width | Single-input | Multi-input(>=2) | |
| GPT-4o | 93.1 | 87.5 | 79.1 | 89.5 | 88.4 | 88.2 | 89.1 | 86.0 | 88.54 |
| LLaVA-1.5-13B | 71.8 | 67.2 | 46.7 | 69.9 | 63.7 | 63.6 | 67.3 | 53.8 | 64.81 |
| DeepSeek-VL-7B | 74.6 | 71.4 | 55.2 | 73.4 | 71.6 | 64.7 | 71.2 | 60.5 | 69.27 |
| Phi-3.5-Vision-4B | 74.0 | 66.1 | 47.0 | 68.8 | 66.1 | 64.2 | 67.7 | 57.6 | 65.85 |
| Qwen2.5-VL-7B | 86.4 | 77.1 | 67.9 | 83.5 | 81.2 | 76.9 | 81.2 | 74.4 | 79.97 |
| - Provision | 87.6 | 79.2 | 70.8 | 84.7 | 83.4 | 78.6 | 83.4 | 74.5 | 81.78 |
| - VLB | 88.9 | 80.6 | 73.4 | 85.6 | 84.9 | 80.7 | 85.4 | 74.5 | 83.41 |
| - **SCOPE** | **92.8** | **84.7** | **76.7** | **89.7** | **87.0** | **86.2** | **89.3** | **77.7** | **87.20** |
| InternVL-2.5-2B | 83.1 | 76.7 | 60.1 | 79.3 | 77.0 | 73.9 | 77.8 | 69.1 | 76.22 |
| - Provision | 83.5 | 77.3 | 63.6 | 79.9 | 79.1 | 74.4 | 79.0 | 70.2 | 77.41 |
| - VLB | 84.8 | 78.9 | 68.5 | 82.1 | 83.3 | 74.4 | 81.7 | 70.4 | 79.63 |
| - **SCOPE** | **87.6** | **80.7** | **70.5** | **85.2** | **83.9** | **78.5** | **83.7** | **74.3** | **82.03** |

**Human acceptance analysis.** To assess the correctness and coherence of the *SCOPE-Test* instances, we conducted human evaluations across three successive rounds of expansion. For each round, 1,000 randomly sampled instances were reviewed to verify the alignment between program outputs and target answers. The resulting human acceptance rates were 96%, 92%, and 91%, respectively, confirming that **SCOPE** generates logically valid and progressively challenging instances.

**Evaluation metric and open-source evolution.** We use accuracy as the evaluation metric, computed along the three evolution dimensions—difficulty, dependency structure, and input modality—to assess model performance under diverse expansion types. To promote transparency and reproducibility, we release the full **SCOPE** benchmark along with tools for program-based, controllable dataset expansion. More importantly, **SCOPE** is designed as a generalizable framework rather than a fixed dataset: we encourage researchers to apply it across domains to evolve their own multimodal benchmarks. By enabling community-driven, multi-round dataset construction, **SCOPE** provides a foundation for an open, collaborative ecosystem of benchmark development and evaluation.

# 5 Experiment

To evaluate the **SCOPE** methodology and its benchmark, we conduct a series of comprehensive experiments. Section 5.1 outlines the experimental setup. We then present three main evaluations: (1) *SCOPE-Test* enables fine-grained diagnosis of existing models across multiple reasoning dimensions (Section 5.2); (2) *SCOPE-Train* improves reasoning performance across models with different parameter scales (Section 5.3); and (3) MAP's staged learning process yields notable gains in visual dialog tasks requiring sequential, context-aware reasoning (Section 5.4). Finally, we conduct ablation studies to assess the contributions of each component in our framework (Section 5.5).

## 5.1 Experimental Setup

**Model Setup.** We train multimodal models using *SCOPE-Train* on two architectures of different scales: Qwen2.5-VL-7B [3] and InternVL-2.5-2B [7]. For program generation, we use GPT-4o [22], a multimodal language model, to produce executable Python code. Unlike ViperGPT [39], which targets end-to-end VQA, **SCOPE** focuses on dataset expansion and evolution. To support this, both the input image and ground-truth answer are provided as prompts to guide more accurate, task-aligned program synthesis.

Table 3: Comparison of various models and methods on different benchmarks.

| Model | MMVet$_{turbo}$ | RealWorldQA | MMBench$_{test}$ | SeedBench2-Plus | MuirBench | MMT$_{val}$ | POPE$_{avg}$ |
|---|---|---|---|---|---|---|---|
| GPT-4o | 69.1 | 75.4 | 83.4 | 72.0 | 68.0 | 65.4 | 86.9 |
| LLaVA-1.5-13B | 38.3 | 55.3 | 64.3 | - | 24.4 | 52.1 | 85.9 |
| DeepSeek-VL-7B | 41.5 | 54.2 | 73.2 | - | - | 54.0 | 88.1 |
| Phi-3.5-Vision-4B | 43.2 | 53.6 | 76.0 | 62.2 | - | 53.6 | - |
| Qwen2.5-VL-7B | 67.1 | 68.5 | 83.5 | 70.4 | 59.6 | 60.9 | 86.4 |
| - Provision | 66.4 | 69.1 | 82.7 | 70.1 | 59.8 | 59.3 | 85.6 |
| - VLB | 66.8 | 69.0 | 84.1 | 70.7 | 61.3 | 61.4 | 84.8 |
| - **SCOPE** | **69.1** | **69.5** | **86.8** | **72.3** | **65.2** | **62.6** | **87.2** |
| InternVL-2.5-2B | 60.8 | 60.1 | 74.7 | 60.9 | 40.6 | 54.5 | 90.6 |
| - Provision | 60.5 | 61.0 | 73.3 | 61.4 | 39.7 | 52.6 | 89.3 |
| - VLB | 60.8 | 60.7 | 75.4 | 61.4 | 40.9 | **55.7** | 90.5 |
| - **SCOPE** | **63.4** | **63.0** | **77.0** | **62.1** | **44.3** | 55.2 | **90.7** |

**Settings.** To ensure fair and comprehensive comparison, we evaluate **SCOPE** against three representative evolution strategies:

*Provision* [45]: This method parses images into scene graphs and extracts subgraphs by selecting specific nodes and edges, thereby generating new instances while preserving relational structure.

*VLB* [42]: This strategy rewrites questions and edits images through predefined templates, recombining them to synthesize new instances without altering the original answers.

**SCOPE** *(Ours)*: Our method constructs executable programs from original samples and systematically expands them along functional composition, input modality, and reasoning complexity, enabling scalable, unbiased, and controllable dataset evolution for multimodal benchmarks.

**Baseline and Benchmark.** To further assess the performance and generalizability of **SCOPE**, we include several strong multimodal baselines spanning different model scales: GPT-4o [22], LLaVA-1.5-13B [29], DeepSeek-VL-7B [32], and Phi-3.5-Vision-4B [1]. For evaluation, we adopt a diverse suite of benchmarks covering both general-purpose and domain-specific tasks: MMVet [44], Real-WorldQA [8], MMBench [31] for broad visual reasoning and real-world contextual understanding, Seedbench2-Plus [26] for text-rich visual comprehension, MMT [43], MuirBench [40] for multi-image inference, and POPE [27] for robustness against hallucination. Notably, the MMBench test set is not public, and SeedBench2-plus does not overlap with SeedBench2. Our expanded data is distinct from all test benchmarks, ensuring no risk of data leakage. We additionally use ConvBench [30], ContextVD [46], and VisDial [9] to assess Visual Dialog capabilities.

## 5.2 MLLM Evaluation on *SCOPE-Test*

We evaluate existing multimodal models on *SCOPE-Test*, focusing on performance across the three structured evolution dimensions. Results are summarized in Table 2, and key observations are as follows. **Reasoning Difficulty.** Model performance degrades substantially on hard instances, which typically require multi-hop reasoning or the integration of multiple intermediate results. This indicates persistent limitations in logical composition and complex inference across current MLLMs. **Structural Variation.** Models perform better on depth-oriented tasks, where each step builds sequentially on previous outputs, than on width-oriented tasks requiring simultaneous attention to multiple variables. This may reflect the influence of prevalent CoT tuning, which favors linear reasoning but offers limited support for managing concurrent dependencies or irrelevant distractions. **Source Modality.** Multi-image reasoning, especially across domains, remains a key challenge. GPT-4o shows the strongest cross-domain performance, underscoring the gap between proprietary models and current open-source MLLMs in broad multimodal generalization. These findings collectively underscore the need for benchmarks like **SCOPE** that systematically expose fine-grained reasoning challenges beyond conventional evaluations.

## 5.3 *SCOPE-Train* Enhances Model Reasoning Capabilities

To assess the effectiveness of the **SCOPE** expansion strategy, we perform a comparative study using the same initial dataset augmented through different baseline methods. Results are summarized in Table 3, leading to the following key observations. **Structured expansions consistently improve performance.** Compared to baseline, all three structured evolution methods—Provision, VLB, and

Table 4: Evaluation on the visual dialog task.

| | ConvBench $R_2$ | ContextVD Acc | VisDial$_{val}$ R@1 |
|---|---|---|---|
| Qwen2.5-VL-7B | 41.5 | 87.4 | 76.8 |
| - Provision | 42.6 | 90.5 | 79.3 |
| - VLB | 43.9 | 91.0 | 82.1 |
| - **SCOPE** | **45.2** | **94.3** | **85.6** |
| InternVL-2.5-2B | 39.8 | 85.7 | 74.2 |
| - Provision | 41.7 | 89.9 | 78.0 |
| - VLB | 42.1 | 91.1 | 80.5 |
| - **SCOPE** | **44.8** | **93.6** | **81.8** |

Table 5: Ablation study on MAP.

| | | MMBench$_{test}$ Acc | ContextVD Acc |
|---|---|---|---|
| 0 | InternVL-2.5-2B (**SCOPE**) | 77.0 | 93.6 |
| 1 | *w/o* curriculum scheduler | 76.9 | 93.1 |
| 2 | *w/o* gating module | 75.3 | 87.4 |
| 3 | *w/o* difficulty-aware loss | 75.8 | 91.2 |
| 4 | *w/o* MAP (direct fine-tuning) | 75.1 | 87.1 |
| 5 | Backbone | 74.7 | 85.7 |

**SCOPE**—achieve notable performance gains by enhancing the quality and diversity of training instances. This underscores the critical role of augmentation in strengthening model reasoning. **SCOPE outperforms other structured methods. SCOPE** consistently achieves the highest improvements across all evaluated benchmarks. Its advantage stems from broader data coverage, unconstrained by scene-graph subgraph limitations or fixed-answer rewrites. Moreover, **SCOPE** substantially enhances performance on multi-image tasks and reduces hallucination errors, indicating stronger generalization and factual consistency. **Scaling effects are more pronounced in larger models.** The performance gains from *SCOPE-Train* are more significant for larger models (e.g., 7B) compared to smaller ones (e.g., 2B). This trend likely reflects the increased benefit derived from **SCOPE**'s multi-round expansion, which introduces a higher proportion of complex, high-difficulty instances that better align with the capacities of larger models. These findings show that *SCOPE-Train* enhances generalization across domains and scales with model capacity, supporting its utility for robust multimodal reasoning.

## 5.4 Evaluation on Visual Dialog Task

To further evaluate the advantages of *SCOPE-Train* and its curriculum learning strategy, we investigate their effectiveness in the context of Visual Dialog, a task characterized by sequential reasoning and context-dependent understanding. Unlike conventional augmentation methods, **SCOPE** enables continuous and controllable expansion of each original sample while preserving its core semantics. This design enhances both sample diversity and structural alignment. As shown in Table 4, models trained with **SCOPE** exhibit superior performance in multi-turn dialog reasoning tasks, particularly those requiring contextual consistency and progressive understanding. We attribute this improvement to the staged learning process: for each training instance, the model incrementally encounters three increasingly challenging expansions, allowing it to gradually build the capacity needed for handling dialog-based visual reasoning with greater fluency and robustness.

## 5.5 Ablation Experiment

**Analysis of SCOPE extension method.** To analyze the contribution of each expansion strategy to model performance, we conduct controlled ablation experiments by incrementally applying different combinations of expansion methods (R-Reasoning Path Expansion, V-Visual Context Editing, C-Cross-Instance Composition) to the same backbone, using the original dataset as a baseline. As shown in Table 6, our structured evolution strategies consistently improve model accuracy, with reasoning path expansion yielding the most notable individual gains. Furthermore, we observe clear synergistic effects when combining multiple expansion types, demonstrating that multi-dimensional augmentation provides complementary benefits beyond any single method.

**Analysis on the MAP.** To evaluate the effectiveness of each component within the MAP curriculum learning framework, we conduct ablation studies by individually removing each module and observing its impact on final model performance (Table 5). Results show that eliminating the curriculum scheduler leads to the smallest performance drop, suggesting that while easy-to-hard progression aids gradual learning, it does not alter the overall quantity or quality of training samples. In contrast, removing either the memory and attention modules or the difficulty-aware loss results in substantial performance degradation, indicating that these components are essential for fully exploiting the structured advantages of **SCOPE**'s expanded data. Furthermore, we additionally include a w/o MAP setting that directly fine-tunes the backbone model. The noticeable performance gap between this variant and the full MAP configuration confirms that MAP itself contributes complementary gains

|   |            | Label |   |   | Accuracy | |
|---|------------|---|---|---|-----------------|----------|
|   |            | R | V | C | MMBench$_{test}$ | ContextVD |
| 0 | Backbone   |   |   |   | 74.7 | 85.7 |
| 1 | + R        | ✓ |   |   | 75.6 | 89.0 |
| 2 | + V        |   | ✓ |   | 75.3 | 88.2 |
| 3 | + C        |   |   | ✓ | 74.9 | 88.5 |
| 4 | + R + V    | ✓ | ✓ |   | 76.4 | 91.4 |
| 5 | + R + C    | ✓ |   | ✓ | 76.1 | 91.3 |
| 6 | + V + C    |   | ✓ | ✓ | 75.8 | 89.7 |
| 7 | InternVL-2.5-2B (**SCOPE**) | ✓ | ✓ | ✓ | 77.0 | 93.6 |

Table 6: Ablation study on extended data.

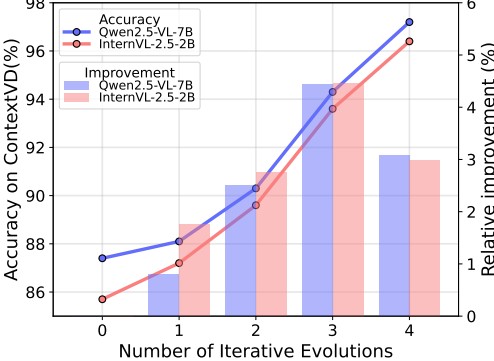

Figure 6: Ablation study on extended iterations.

beyond data quality. This extended analysis helps disentangle the respective effects of the dataset and the training strategy, providing a clearer picture of MAP's independent impact.

**Analysis on the number of evolutions.** We analyze model performance after each round of iterative expansion, as shown in the line plot, and focus on the relative improvement between consecutive rounds using bar plots to control for the impact of increased data volume (Figure 6). The results reveal consistent performance gains across expansion rounds, with relative improvements stabilizing after the third iteration. While further expansion could continue to enhance model performance, we limit the number of rounds to three due to computational cost considerations.

# 6 Discussion: Generalizability of Visual Programming

A key advantage of adopting the visual programming paradigm is its broad generalizability beyond conventional VQA. Existing program-based frameworks such as ViperGPT [39], VISPROG [17], and De-fine [15] have already demonstrated that modular, Python-like programs can perform not only image QA but also visual grounding, instruction following, image editing, and even video understanding—all without retraining or task-specific architectures. By representing multimodal reasoning as executable code, **SCOPE** provides a unified and interpretable interface that can be adapted to a wide range of visual tasks.

Building on this foundation, **SCOPE**'s evolution process operates purely at the program level, meaning that the same principles of reasoning-path expansion, visual context editing, and cross-instance composition can be applied to any task where outputs can be represented as code execution results. In practice, this includes tasks such as instruction-conditioned image generation, cross-frame video reasoning, and visual planning. Moreover, the modular API design allows seamless extension: new domain-specific functions can be incorporated into the library without redesigning templates or retraining the model, enabling scalable adaptation to emerging domains.

In summary, the visual programming paradigm empowers **SCOPE** to serve as a general, interpretable, and extensible framework for multimodal reasoning and generation, capable of producing structured code outputs for virtually any vision-language task.

# 7 Conclusion

In conclusion, we present **SCOPE**, a program-centric framework for controllable and verifiable dataset evolution in multimodal learning. By leveraging visual programming, counterfactual reasoning, and the MAP curriculum, **SCOPE** enables systematic data expansion with explicit reasoning paths and adaptive difficulty control. Our constructed benchmarks, *SCOPE-Train* and *SCOPE-Test*, not only reveal critical limitations in existing models but also promote scalable and interpretable training, paving the way for more robust and generalizable MLLMs.

## 8 Acknowledgement

This work was supported by the National Key Research and Development Program of China (2024YFB3312900), the National Natural Science Foundation of China (62441617), Zhejiang Provincial Natural Science Foundation of China (No. LD25F020001) and Fundamental Research Funds for the Central Universities (226-2025-00057), Ningbo Yongjiang Talent Introduction Programme (2024A-401-G), the Zhejiang NSF (LRG25F020001), Zhejiang University Education Foundation Qizhen Scholar Foundation, Wallenberg-NTU Presidential Postdoctoral Fellowship.

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

# 9 Appendix

This is the Appendix for the paper "Counterfactual Evolution of Multimodal Datasets via Visual Programming". We organize the appendices and supplementary information mentioned in the paper as follows:

- 7.1 SCOPE Raw Data Distribution;
- 7.2 Detailed Distribution of SCOPE after Expansion;
- 7.3 A SCOPE Data Example in Annotation;
- 7.4 The Corresponding Picture in Figure 4;
- 7.5 Limitation;
- 7.6 API and Function Library in Program Generation

## 9.1 SCOPE Raw Data Distribution

We selected six publicly available datasets as the initial data sources for SCOPE evolution. The specific data distributions and selected subsets are detailed in Table 6.

Table 7: Data mixture of VQA benchmarks and datasets.

| Dataset | Set | Description | Number of samples |
|---|---|---|---|
| Seedbench2 | | General Benchmark | 5543 |
| MMBench | | General Benchmark | 2521 |
| MME | | General Benchmark | 2042 |
| GQA | test-dev | Compositional | 1426 |
| OK-VQA | val | External Knowledge | 1186 |
| TallyQA | test | Counting | 2156 |
| Total | | | 14874 |

## 9.2 Detailed Distribution of SCOPE after Expansion

SCOPE underwent three rounds of evolution in total. The data distributions for the first two rounds are additionally provided in Figures 7 and 8.

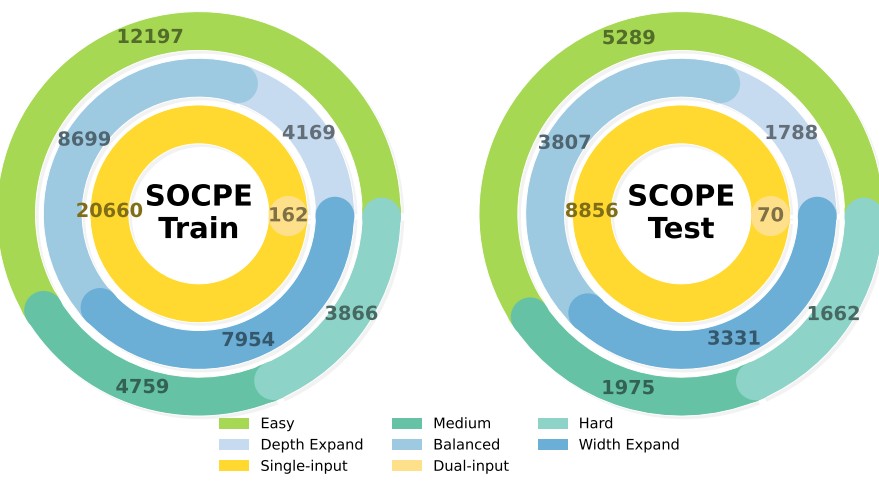

Figure 7: SCOPE dataset after two evolutions

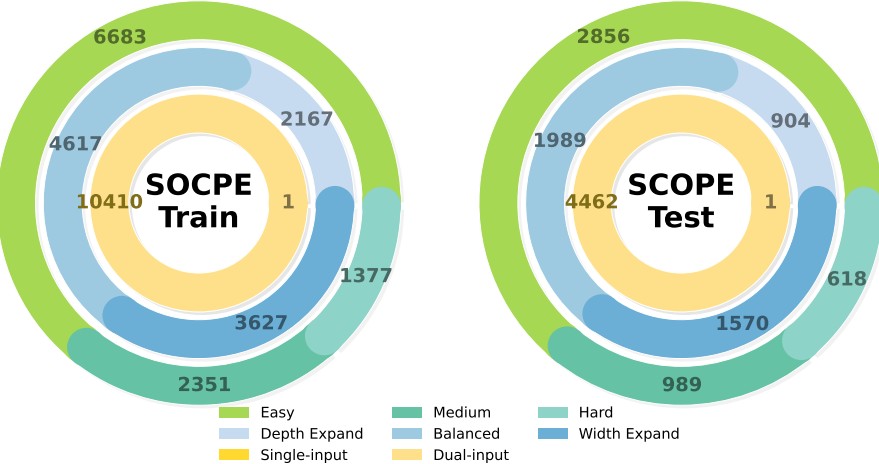

Figure 8: SCOPE dataset after only one evolution

## 9.3 A SCOPE Data Example in Annotation

We present a concrete annotation example, which is followed by both the SCOPE-Train and SCOPE-Test datasets.

```json
{
  "origin_data": {
    "data_source": "MME",
    "source_id": "856",
    "question": "Will green be obtained by mixing the above two colors? Please answer yes or
      no.",
    "image": [
      "images/MME/test1_856.jpg"
    ],
    "golden_answer": [
      "Yes"
    ],
    "program": "def execute_command_856(image, possible_answers, query, ImagePatch,
      VideoSegment, llm_query, bool_to_yesno, distance, best_image_match):\n     # Answer is:def
      execute_command(image, my_fig, time_wait_between_lines, syntax):\n     image_patch =
      ImagePatch(image)\n    \n    # Find the two colors in the image\n    color_patches =
      image_patch.find(\"color\")\n    show_all(None, color_patches, \"color_patches\")\n    \n
          if len(color_patches) < 2:\n         return \"no\"  # Not enough colors to mix\n    \n
          # Get the names of the two colors\n    color_1 = color_patches[0].simple_query(\"What
      is the name of this color?\")\n    show_all(None, color_1, \"color_1\")\n    color_2 =
      color_patches[1].simple_query(\"What is the name of this color?\")\n    show_all(None,
      color_2, \"color_2\")\n    \n    # Use llm_query to determine if mixing the two colors
      results in green\n    query = f\"Will mixing {color_1} and {color_2} result in green?
      Answer yes or no.\"\n    result = image_patch.llm_query(query, long_answer=False)\n
      show_all(None, result, \"result\")\n    \n    return result",
    "program_answer": "yes",
    "static analysis": {
      "difficulty": "easy",
      "extend_method": null
    },
    "image_count": 1
  },
  "evolution_data_1": {
    "question": "Will green be obtained by mixing the above two colors? Please answer yes or
      no.",
    "image": [
      "images/MME/mme_sample_856.jpg"
    ],
    "program": "def execute_command_856(image, possible_answers, query, ImagePatch,
      VideoSegment, llm_query, bool_to_yesno, distance, best_image_match):\n     # Answer is:def
      execute_command(image, my_fig, time_wait_between_lines, syntax):\n     image_patch =
      ImagePatch(image)\n    \n    # Find the two colors in the image\n    color_patches =
      image_patch.find(\"color\")\n    show_all(None, color_patches, \"color_patches\")\n    \n
          # Ensure there are at least two colors to mix\n    if len(color_patches) < 2:\n
          return \"no\"\n    \n    # Get the names of the two colors\n    color_1 = color_patches
      [0].simple_query(\"What is the name of this color?\")\n    show_all(None, color_1, \"
      color_1\")\n    color_2 = color_patches[1].simple_query(\"What is the name of this color
```

```
                ?\")\n      show_all(None, color_2, \"color_2\")\n      \n      # Use llm_query to determine if
           mixing the two colors results in green\n      query = f\"Will mixing {color_1} and {
           color_2} result in green? Answer yes or no.\"\n      result = image_patch.llm_query(query,
           long_answer=False)\n      show_all(None, result, \"result\")\n      \n      return result",
26      "program_answer": "Yes",
27      "static analysis": {
28        "difficulty": "easy",
29        "extend_method": "balanced"
30      },
31      "image_count": 1
32    },
33    "evolution_data_2": {
34      "question": "If green is obtained by mixing the two colors, which color is more dominant
           in the resulting green?",
35      "image": [
36        "images/MME/mme_sample_856.jpg"
37      ],
38      "program": "def execute_command_856(image, possible_answers, query, ImagePatch,
           VideoSegment, llm_query, bool_to_yesno, distance, best_image_match):\n    # Answer is:def
           execute_command(image):\n    colors = []\n    for img in image:\n        patch =
           ImagePatch(img)\n        color = patch.simple_query(\"What is the color?\")\n
           colors.append(color.lower())\n    \n    if \"yellow\" in colors and \"blue\" in colors:\n
           yellow_patch = None\n        blue_patch = None\n        for img in image:\n
           patch = ImagePatch(img)\n            if patch.simple_query(\"What is the color?\").
           lower() == \"yellow\":\n                yellow_patch = patch\n            elif patch.
           simple_query(\"What is the color?\").lower() == \"blue\":\n                blue_patch =
           patch\n        \n        if yellow_patch and blue_patch:\n            yellow_area =
           yellow_patch.width * yellow_patch.height\n            blue_area = blue_patch.width *
           blue_patch.height\n            return \"yellow\" if yellow_area > blue_area else \"blue\"\
           n    return \"no green obtained\"",
39      "program_answer": "no green obtained",
40      "static analysis": {
41        "difficulty": "hard",
42        "extend_method": "width"
43      },
44      "image_count": 1
45    },
46    "evolution_data_3": {
47      "question": "If green is obtained by mixing yellow and blue, which color contributes more
           to the green in terms of spatial dominance across all images, and how does their spatial
           relationship (left/right) affect the resulting green?",
48      "image": [
49        "images/MME/extended_data_2_MME_commonsense_reasoning_0009.jpg",
50        "images/MME/mme_sample_856.jpg"
51      ],
52      "program": "def execute_command_856(image, possible_answers, query, ImagePatch,
           VideoSegment, llm_query, bool_to_yesno, distance, best_image_match):\n    # Answer is:def
           execute_command(image):\n    dominant_colors = []\n    spatial_relationships = []\n    \
           n    for img in image:\n        patch = ImagePatch(img)\n        yellow_patch = patch.
           find(\"yellow\")\n        blue_patch = patch.find(\"blue\")\n        \n        if
           yellow_patch and blue_patch:\n            yellow_area = yellow_patch[0].width *
           yellow_patch[0].height\n            blue_area = blue_patch[0].width * blue_patch[0].
           height\n            if yellow_area > blue_area:\n
           dominant_colors.append(\"yellow\")\n            else:\n                dominant_colors.
           append(\"blue\")\n            \n            # Determine spatial relationship\n
           if yellow_patch[0].horizontal_center < blue_patch[0].horizontal_center:\n
           spatial_relationships.append(\"yellow is left of blue\")\n            else:\n
           spatial_relationships.append(\"blue is left of yellow\")\n    \n    # Determine the
           overall dominant color across all images\n    yellow_count = dominant_colors.count(\"
           yellow\")\n    blue_count = dominant_colors.count(\"blue\")\n    \n    dominant_color =
           \"yellow\" if yellow_count > blue_count else \"blue\"\n    \n    # Determine the most
           common spatial relationship\n    spatial_relationship = max(set(spatial_relationships),
           key=spatial_relationships.count)\n    \n    return f\"The dominant color is {
           dominant_color}, and {spatial_relationship}.\"",
53      "program_answer": "The dominant color is yellow, and yellow is left of blue.",
54      "static analysis": {
55        "difficulty": "hard",
56        "extend_method": "balanced"
57      },
58      "image_count": 2
59  }
60}
```

Listing 1: Annotation example.

## 9.4 The Corresponding Picture in Figure 4

In Figure 4, we illustrate the process of Counterfactual Categorization from a code-level perspective. Due to space limitations, the corresponding visual examples are provided in the appendix.

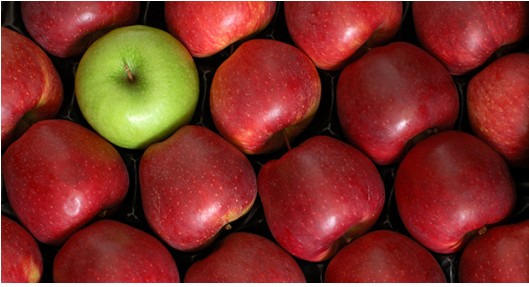

Figure 9: Original image

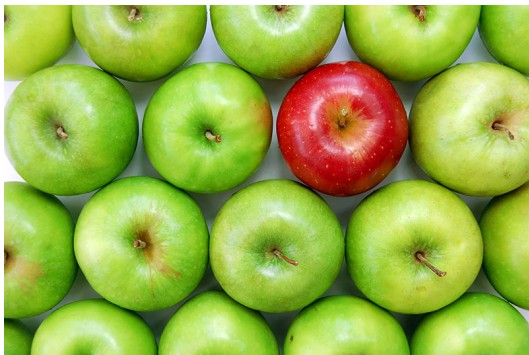

Figure 10: Associated image through Cross-Instance Composition.

## 9.5 Limitation

In this section, we discuss several open directions and current limitations for future exploration. First, although our dataset construction relies on six publicly available datasets, SCOPE is designed as a dataset-agnostic evolution algorithm and can be applied to data from various domains. We encourage future work to extend and validate its effectiveness on proprietary datasets. Second, in the MAP curriculum learning stage, the difficulty ranking of structural variation is primarily based on heuristic insights. We remain open to refining this ordering should more principled or effective strategies become available.

## 9.6 API and Function Library in Program Generation

The prompts employed during the code generation process are listed in detail.

```
import math

class ImagePatch:
    """A Python class containing a crop of an image centered around a particular object, as
     well as relevant information.
    Attributes
    ----------
    cropped_image : array_like
        An array-like of the cropped image taken from the original image.
    left, lower, right, upper : int
        An int describing the position of the (left/lower/right/upper) border of the crop's
     bounding box in the original image.

    Methods
    -------
    find(object_name: str)->List[ImagePatch]
```

```
15            Returns a list of new ImagePatch objects containing crops of the image centered around
          any objects found in the
16            image matching the object_name.
17        exists(object_name: str)->bool
18            Returns True if the object specified by object_name is found in the image, and False
          otherwise.
19        verify_property(property: str)->bool
20            Returns True if the property is met, and False otherwise.
21        best_text_match(option_list: List[str], prefix: str)->str
22            Returns the string that best matches the image.
23        simple_query(question: str=None)->str
24            Returns the answer to a basic question asked about the image. If no question is
          provided, returns the answer to "What is this?".
25        compute_depth()->float
26            Returns the median depth of the image crop.
27        crop(left: int, lower: int, right: int, upper: int)->ImagePatch
28            Returns a new ImagePatch object containing a crop of the image at the given
          coordinates.
29        """

31        def __init__(self, image, left: int = None, lower: int = None, right: int = None, upper:
          int = None):
32            """Initializes an ImagePatch object by cropping the image at the given coordinates and
          stores the coordinates as
33            attributes. If no coordinates are provided, the image is left unmodified, and the
          coordinates are set to the
34            dimensions of the image.
35            Parameters
36            -------
37            image : array_like
38                An array-like of the original image.
39            left, lower, right, upper : int
40                An int describing the position of the (left/lower/right/upper) border of the crop'
          s bounding box in the original image.
41            """
42            if left is None and right is None and upper is None and lower is None:
43                self.cropped_image = image
44                self.left = 0
45                self.lower = 0
46                self.right = image.shape[2]  # width
47                self.upper = image.shape[1]  # height
48            else:
49                self.cropped_image = image[:, lower:upper, left:right]
50                self.left = left
51                self.upper = upper
52                self.right = right
53                self.lower = lower

55            self.width = self.cropped_image.shape[2]
56            self.height = self.cropped_image.shape[1]

58            self.horizontal_center = (self.left + self.right) / 2
59            self.vertical_center = (self.lower + self.upper) / 2

61        def find(self, object_name: str) -> List[ImagePatch]:
62            """Returns a list of ImagePatch objects matching object_name contained in the crop if
          any are found.
63            Otherwise, returns an empty list.
64            Parameters
65            ----------
66            object_name : str
67                the name of the object to be found

69            Returns
70            -------
71            List[ImagePatch]
72                a list of ImagePatch objects matching object_name contained in the crop

74            Examples
75            --------
76            >>> # return the foo
77            >>> def execute_command(image) -> List[ImagePatch]:
78            >>>     image_patch = ImagePatch(image)
79            >>>     foo_patches = image_patch.find("foo")
80            >>>     return foo_patches
81            """
82            return find_in_image(self.cropped_image, object_name)

84        def exists(self, object_name: str) -> bool:
85            """Returns True if the object specified by object_name is found in the image, and
          False otherwise.
```

```
86          Parameters
87          -------
88          object_name : str
89              A string describing the name of the object to be found in the image.
90
91          Examples
92          -------
93          >>> # Are there both foos and garply bars in the photo?
94          >>> def execute_command(image)->str:
95          >>>     image_patch = ImagePatch(image)
96          >>>     is_foo = image_patch.exists("foo")
97          >>>     is_garply_bar = image_patch.exists("garply bar")
98          >>>     return bool_to_yesno(is_foo and is_garply_bar)
99          """
100         return len(self.find(object_name)) > 0
101
102     def verify_property(self, object_name: str, visual_property: str) -> bool:
103         """Returns True if the object possesses the visual property, and False otherwise.
104         Differs from 'exists' in that it presupposes the existence of the object specified by
        object_name, instead checking whether the object possesses the property.
105         Parameters
106         -------
107         object_name : str
108             A string describing the name of the object to be found in the image.
109         visual_property : str
110             A string describing the simple visual property (e.g., color, shape, material) to
        be checked.
111
112         Examples
113         -------
114         >>> # Do the letters have blue color?
115         >>> def execute_command(image) -> str:
116         >>>     image_patch = ImagePatch(image)
117         >>>     letters_patches = image_patch.find("letters")
118         >>>     # Question assumes only one letter patch
119         >>>     return bool_to_yesno(letters_patches[0].verify_property("letters", "blue"))
120         """
121         return verify_property(self.cropped_image, object_name, property)
122
123     def best_text_match(self, option_list: List[str], prefix: str=None) -> str:
124         """Returns the string that best matches the image.
125         Parameters
126         -------
127         option_list : str
128             A list with the names of the different options
129         prefix : str
130             A string with the prefixes to append to the options
131
132         Examples
133         -------
134         >>> # Is the foo gold or white?
135         >>> def execute_command(image)->str:
136         >>>     image_patch = ImagePatch(image)
137         >>>     foo_patches = image_patch.find("foo")
138         >>>     # Question assumes one foo patch
139         >>>     return foo_patches[0].best_text_match(["gold", "white"])
140         """
141         return best_text_match(self.cropped_image, option_list, prefix)
142
143     def simple_query(self, question: str = None) -> str:
144         """Returns the answer to a basic question asked about the image. If no question is
        provided, returns the answer
145         to "What is this?". The questions are about basic perception, and are not meant to be
        used for complex reasoning
146         or external knowledge.
147         Parameters
148         -------
149         question : str
150             A string describing the question to be asked.
151
152         Examples
153         -------
154
155         >>> # Which kind of baz is not fredding?
156         >>> def execute_command(image) -> str:
157         >>>     image_patch = ImagePatch(image)
158         >>>     baz_patches = image_patch.find("baz")
159         >>>     for baz_patch in baz_patches:
160         >>>         if not baz_patch.verify_property("baz", "fredding"):
161         >>>             return baz_patch.simple_query("What is this baz?")
162
```

```python
163         >>> # What color is the foo?
164         >>> def execute_command(image) -> str:
165         >>>     image_patch = ImagePatch(image)
166         >>>     foo_patches = image_patch.find("foo")
167         >>>     foo_patch = foo_patches[0]
168         >>>     return foo_patch.simple_query("What is the color?")
169
170         >>> # Is the second bar from the left quuxy?
171         >>> def execute_command(image) -> str:
172         >>>     image_patch = ImagePatch(image)
173         >>>     bar_patches = image_patch.find("bar")
174         >>>     bar_patches.sort(key=lambda x: x.horizontal_center)
175         >>>     bar_patch = bar_patches[1]
176         >>>     return bar_patch.simple_query("Is the bar quuxy?")
177         """
178         return simple_query(self.cropped_image, question)
179
180     def compute_depth(self):
181         """Returns the median depth of the image crop
182         Parameters
183         ----------
184         Returns
185         -------
186         float
187             the median depth of the image crop
188
189         Examples
190         --------
191         >>> # the bar furthest away
192         >>> def execute_command(image)->ImagePatch:
193         >>>     image_patch = ImagePatch(image)
194         >>>     bar_patches = image_patch.find("bar")
195         >>>     bar_patches.sort(key=lambda bar: bar.compute_depth())
196         >>>     return bar_patches[-1]
197         """
198         depth_map = compute_depth(self.cropped_image)
199         return depth_map.median()
200
201     def crop(self, left: int, lower: int, right: int, upper: int) -> ImagePatch:
202         """Returns a new ImagePatch cropped from the current ImagePatch.
203         Parameters
204         -------
205         left, lower, right, upper : int
206             The (left/lower/right/upper)most pixel of the cropped image.
207         -------
208         """
209         return ImagePatch(self.cropped_image, left, lower, right, upper)
210
211     def overlaps_with(self, left, lower, right, upper):
212         """Returns True if a crop with the given coordinates overlaps with this one,
213         else False.
214         Parameters
215         ----------
216         left, lower, right, upper : int
217             the (left/lower/right/upper) border of the crop to be checked
218
219         Returns
220         -------
221         bool
222             True if a crop with the given coordinates overlaps with this one, else False
223
224         Examples
225         --------
226         >>> # black foo on top of the qux
227         >>> def execute_command(image) -> ImagePatch:
228         >>>     image_patch = ImagePatch(image)
229         >>>     qux_patches = image_patch.find("qux")
230         >>>     qux_patch = qux_patches[0]
231         >>>     foo_patches = image_patch.find("black foo")
232         >>>     for foo in foo_patches:
233         >>>         if foo.vertical_center > qux_patch.vertical_center:
234         >>>             return foo
235         """
236         return self.left <= right and self.right >= left and self.lower <= upper and self.
    upper >= lower
237
238 def best_image_match(list_patches: List[ImagePatch], content: List[str], return_index=False)
       -> Union[ImagePatch, int]:
239     """Returns the patch most likely to contain the content.
240     Parameters
241     ----------
```

```python
242    list_patches : List[ImagePatch]
243    content : List[str]
244        the object of interest
245    return_index : bool
246        if True, returns the index of the patch most likely to contain the object
247
248    Returns
249    -------
250    int
251        Patch most likely to contain the object
252    """
253    return best_image_match(list_patches, content, return_index)

256 def distance(patch_a: ImagePatch, patch_b: ImagePatch) -> float:
257    """
258    Returns the distance between the edges of two ImagePatches. If the patches overlap, it
         returns a negative distance
259    corresponding to the negative intersection over union.
260
261    Parameters
262    ----------
263    patch_a : ImagePatch
264    patch_b : ImagePatch
265
266    Examples
267    --------
268    # Return the qux that is closest to the foo
269    >>> def execute_command(image):
270    >>>     image_patch = ImagePatch(image)
271    >>>     qux_patches = image_patch.find('qux')
272    >>>     foo_patches = image_patch.find('foo')
273    >>>     foo_patch = foo_patches[0]
274    >>>     qux_patches.sort(key=lambda x: distance(x, foo_patch))
275    >>>     return qux_patches[0]
276    """
277    return distance(patch_a, patch_b)

280 def bool_to_yesno(bool_answer: bool) -> str:
281    return "yes" if bool_answer else "no"

283 def coerce_to_numeric(string):
284    """
285    This function takes a string as input and returns a float after removing any non-numeric
         characters.
286    If the input string contains a range (e.g. "10-15"), it returns the first value in the
         range.
287    """
288    return coerce_to_numeric(string)

290 Write a function using Python and the ImagePatch class (above) that could be executed to
         provide an answer to the query.

292 Consider the following guidelines:
293 - Use base Python (comparison, sorting) for basic logical operations, left/right/up/down, math,
         etc.
294 - Using the simple_query function to ask questions that require image information.
295 - You may be given multiple images as input. Use all of them when answering the question. Each
         image can be accessed separately, like: image[0], image[1].

297 -You are given a list of images as input. Each item in this list is a single image,so Always
         treat the input variable image as a list.
298 -**Very important**, You MUST loop over each image and wrap it with the ImagePatch class:
299    for img in image:
300        patch = ImagePatch(img)

302 When dealing with a list of images, avoid returning inside the for-loop. Instead, collect all
         relevant information across images and return the final answer after the loop.
303 If an object must be verified across multiple images, gather data first, then return a final
         summary:
304 - You **must examine all images**.
305 - However, the final output should be a **single, unified answer**, not a list of per-image
         outputs.
306 - Do not return inside the loop. Instead, gather all relevant information across the images
         and generate one overall result.
307 - Even if some intermediate reasoning is per-image, combine the insights before returning.
308 Bad:
309    return [answer_img1, answer_img2, ...]
310 Good:
```

22

```
311    Analyze all - summarize - return one final answer (e.g., "yes", or an integer, or a
        sentence)
312
313Query: INSERT_QUERY_HERE
314Answer: INSERT_ANSWER_HERE
315---
316
317First, write a Python function to answer the original visual question.
318Then, generate a follow-up question that requires deeper image reasoning-e.g., multi-object
        logic, spatial relations, or attribute comparison.
319Write a second function to solve it, using higher code complexity, lower similarity, and
        involving more of the previously mentioned functions or APIs.
320
321Format your answer as follows:
322Original Code:
323<original_code>
324
325Follow-up Question:
326<followup_question>
327
328Follow-up Code:
329<followup_code>
```

Listing 2: Prompt for code generation.

