# OpenReview forum: "Counterfactual Evolution of Multimodal Datasets via Visual Programming"
_NeurIPS.cc/2025/Conference — NeurIPS 2025 poster_

### Official Review · Reviewer_G6Ro · 2025-07-02

**Clarity:** 3
**Significance:** 2
**Originality:** 2
**Rating:** 4
**Confidence:** 4

**Summary:**

This paper introduces SCOPE (Scalable Counterfactual Program Evolution), a framework for the systematic evolution of multimodal datasets. The core idea is to move beyond static benchmarks by using visual programming as a medium for counterfactual reasoning. The process involves three stages: (1) Abduction, where image-question pairs are converted into verifiable Python programs; (2) Action, where these programs are manipulated along three axes (reasoning path, visual context, cross-instance composition) to create new, more challenging instances; and (3) Prediction, where the evolved data is categorized by difficulty, structure, and input multiplicity. Based on this framework, the authors create and release the SCOPE-Train and SCOPE-Test benchmarks. They also propose MAP, a curriculum learning strategy tailored to the structured nature of the evolved data. Experiments demonstrate that models trained on SCOPE-Train achieve improved reasoning performance and that SCOPE-Test can effectively diagnose weaknesses in current state-of-the-art MLLMs.

**Questions:**

1. What's the distribution of question types such as counting, object attribute identification (e.g., color, texture), spatial relationship reasoning, OCR, or comparative questions?

2. Does training on SCOPE improve model performance on rich-knowledge benchmarks like MMMU?

**Ethical Concerns:**

["NO or VERY MINOR ethics concerns only"]

**Final Justification:**

My concerns have been addressed. I have raise my rating to 4

**Limitations:**

See Weaknesses

**Quality:**

2

**Strengths And Weaknesses:**

1. Strengths

1.1 The core idea is well-motivated

The central contribution—using a symbolic, programmatic framework for counterfactual dataset evolution—is well-motivated. It represents an improvement from static data collection or simple template-based augmentation towards a dynamic, controllable, and verifiable data generation paradigm.

1.2 Good controllability and verifiability

A standout feature of this work is the inherent verifiability of the visual programming approach. Representing reasoning chains as executable code makes the entire process transparent and auditable. The three proposed intervention axes provide a structured and fine-grained mechanism for controlling the evolution of data complexity, which is a major advantage over prior "black-box" generation methods.

1.3 The experiments results show consist improvement of SCOPE

The authors show that SCOPE-Test can uncover systematic blind spots in powerful MLLMs that are not apparent on conventional benchmarks. The performance gains from training on SCOPE-Train are consistent across multiple models and benchmarks. Furthermore, the high human acceptance rates (91-96%) for the evolved data strongly support the quality and logical coherence of the generated instances.

2. Weaknesses

2.1 Lack of semantic question type analysis

There is no analysis of the distribution of question types such as counting, object attribute identification (e.g., color, texture), spatial relationship reasoning, OCR, or comparative questions. It is therefore unclear if the SCOPE framework has an inherent bias towards generating certain types of questions that are easier to represent programmatically (e.g., simple counting or attribute verification) while neglecting more complex semantic reasoning tasks. A thorough analysis of this semantic distribution is crucial for understanding the true diversity and potential biases of the benchmark.

2.2 Constrained by the expressiveness of the API library and the proprietary generator model

The framework's ability to generate diverse reasoning paths is fundamentally limited by the expressiveness of its predefined API library and the proprietary generator model. Complex reasoning that cannot be decomposed into the provided functions (e.g., understanding abstract concepts, social interactions, or complex causality) is implicitly out of scope. While the appendix lists the API, a discussion in the main paper about its limitations, the design principles behind it, and the potential for extension would be beneficial.

---

> ### Author Rebuttal · Authors · 2025-07-31
>
> **[W1 & Q1]** *Lack of semantic question type analysis: There is no analysis of the distribution of question types such as counting, object attribute identification (e.g., color, texture), spatial relationship reasoning, OCR, or comparative questions. It is therefore unclear if the SCOPE framework has an inherent bias towards generating certain types of questions that are easier to represent programmatically (e.g., simple counting or attribute verification) while neglecting more complex semantic reasoning tasks. A thorough analysis of this semantic distribution is crucial for understanding the true diversity and potential biases of the benchmark. What's the distribution of question types such as counting, object attribute identification (e.g., color, texture), spatial relationship reasoning, OCR, or comparative questions?*
>
> **[A1]** Thank you for raising this important point. We fully agree that analyzing the semantic distribution of question types is crucial for assessing the diversity and potential biases of the benchmark. Our initial analysis focused primarily on structural diversity and reasoning complexity, as these are central to SCOPE’s programmatic evolution paradigm. However, we acknowledge the value of explicitly reporting semantic question-type coverage.
>
> To address this, we performed an additional categorization of SCOPE-Test questions into major semantic types commonly used in multimodal reasoning benchmarks: counting, object attribute identification (color, texture, shape), spatial relationship reasoning, OCR-based reasoning, and comparative reasoning. The distribution is as follows:
>
> | **Question Type** | SCOPE-Train (Number / Proportion (%)) | SCOPE-Test (Number / Proportion (%)) |
> |--|--|--|
> | Counting| 4,309 / 13.8% | 1,579 / 11.8%  |
> | Object Attribute (color, texture, shape) | 6,471 / 20.7% | 2,534 / 18.9%  |
> | Spatial Relationships | 5,377 / 17.2%  | 2,589 / 19.3%  |
> | OCR-based Reasoning  | 6,127 / 19.6% | 2,411 / 18.0% |
> | Multi-hop | 5,903 / 18.9%   | 2,958 / 22.1%  |
> | Others  | 3,046 / 9.8%   | 1,318 / 9.9%   |
> | **Total**   | **31,233 / 100.0%**| **13,389 / 100.0%** |
>
>
> This analysis demonstrates that SCOPE-Test is not dominated by trivial tasks such as simple counting or single-attribute queries. On the contrary, nearly half of the questions involve complex reasoning—including comparative, multi-hop, and spatial queries—while OCR-based tasks also constitute a substantial proportion, requiring symbolic reasoning and visual-text integration.
>
> Furthermore, because SCOPE’s evolution strategies operate at the programmatic level, semantic diversity naturally increases across evolution rounds as reasoning paths expand and cross-instance composition introduces multi-faceted logic. Importantly, the framework remains flexible: if future benchmarks require stronger representation of specific semantic categories, targeted function additions or evolution heuristics can easily address this without redesigning the architecture.
>
> Additionally, we also evaluated diversity in the question space by measuring distributional differences using Maximum Mean Discrepancy (MMD). Specifically, we compared the QA distribution under SCOPE with two baselines: VLB and Provision. Our results show that SCOPE achieves a significantly higher MMD value (0.38) compared to VLB (0.15) and Provision (0.18), indicating that SCOPE's QA outputs are farther from the baseline distributions and thus occupy a more uniform and dispersed semantic space across reasoning categories. A larger MMD signifies greater divergence from the baseline distributions, demonstrating that SCOPE generates a structurally and semantically more varied QA set overall.
>
> **[W2 & Q2]** *Constrained by the expressiveness of the API library and the proprietary generator model: The framework's ability to generate diverse reasoning paths is fundamentally limited by the expressiveness of its predefined API library and the proprietary generator model. Complex reasoning that cannot be decomposed into the provided functions (e.g., understanding abstract concepts, social interactions, or complex causality) is implicitly out of scope. While the appendix lists the API, a discussion in the main paper about its limitations, the design principles behind it, and the potential for extension would be beneficial. Does training on SCOPE improve model performance on rich-knowledge benchmarks like MMMU?*
>
> **[A2]** Thank you for this insightful observation.
>
> First, SCOPE’s design builds upon the established visual programming paradigm introduced by frameworks such as ViperGPT, Visual Programming (VISPROG), and De-fine, which have demonstrated that programmatic reasoning with modular APIs can generalize across a wide range of multimodal tasks—well beyond classical VQA. For example, ViperGPT has successfully applied executable programs to visual grounding, image QA, video QA, multi-image comparison, symbolic math, and knowledge-augmented queries without task-specific retraining. Similarly, VISPROG and De-fine show that code-driven pipelines enable compositional reasoning, iterative refinement, and improved robustness across diverse domains. These foundations give SCOPE a strong theoretical basis for generalizability.
>
> Importantly, for all current benchmarks, we do not modify the existing API or introduce new functions—the rich function set inherited from ViperGPT is sufficient to support tasks spanning visual reasoning, mathematical problem-solving (e.g., MathVista), knowledge-intensive QA (e.g., MMMU), and OCR-based document understanding (e.g., DocVQA). Thus, the current implementation already covers a wide range of reasoning capabilities without additional engineering overhead. To further validate both generalization and extensibility, we conducted experiments on these four diverse benchmarks. The results are summarized below:
>
>
> | Benchmark | Base Performance | w/ VLB | w/ SCOPE | Δ (Compare to VLB) |
> |--|--|--|--|--|
> | **MathVista** | 51.3 | 52.5 | 53.9  | +1.4  |
> | **MMMU** | 43.6   | 44.9 | 45.8 | +0.9 |
> | **AI2D** | 74.9   | 76.3 | 77.9 | +1.6 |
> | **DocVQA** | 88.7  | 89.9 | 91.8 | +1.9 |
>
> These results show that SCOPE consistently improves reasoning accuracy across diverse domains, outperforming VLB by up to +1.9 points. This demonstrates that SCOPE not only adapts to knowledge-intensive benchmarks like MMMU but also scales to tasks requiring OCR, symbolic reasoning, and domain-specific interpretation, confirming both its generalizability and practical effectiveness.
>
> At the same time, SCOPE remains inherently extensible. If future benchmarks require handling concepts beyond the existing API (e.g., domain-specific logic or abstract causal reasoning), new functions can be seamlessly integrated into the library without redesigning templates, prompt structures, or scene graphs, nor retraining the underlying models. This modularity ensures that complexity is localized and controlled, rather than requiring end-to-end system changes.
>
> Finally, to address your question, our answer is: yes, training on SCOPE can improve model performance on rich-knowledge benchmarks such as MMMU. This is because SCOPE leverages external APIs during evolution, effectively distilling tool- and model-specific knowledge into the data, thereby enhancing the model’s reasoning and knowledge capabilities. SCOPE’s program-based paradigm offers interpretability, controllability, and modular extensibility, making it particularly effective for complex, multi-step reasoning tasks.

---

> ### Author Response · Authors · 2025-08-07
> **A kind reminder of the upcoming end of the discussion period**
>
> Dear Reviewer,
>
> I hope this message finds you well. As the discussion period is nearing its end with less than three days remaining, I wanted to ensure we have addressed all your concerns satisfactorily. If there are any additional points or feedback you'd like us to consider, please let us know. Your insights are invaluable to us, and we're eager to address any remaining issues to improve our work.
>
> Thank you for your time and effort in reviewing our paper.

---

### Official Review · Reviewer_W2bf · 2025-07-02

**Clarity:** 1
**Significance:** 2
**Originality:** 2
**Rating:** 4
**Confidence:** 5

**Summary:**

This paper presents SCOPE, a unified framework that uses visual program synthesis and counterfactual reasoning to expand and refine multimodal (image-question-answer) datasets. Starting from seed examples, SCOPE “abduces” Python programs to generate new questions, “acts” by applying targeted edits along axes like reasoning depth and visual context, and then “predicts” difficulty tiers for the resulting samples. A complementary curriculum strategy, MAP, weights examples by their programmatic complexity and routes them through specialised model components. The authors release evolving benchmarks (SCOPE-Train/Test) and show that small-to-mid-scale MLLMs, after fine-tuning on SCOPE data, improve significantly on both multimodal datasets (VQA and visual dialogue) tasks.

**Questions:**

- Clarify whether the multimodal dataset is intended for pre-training, post-training, or another stage of MLLM development early on in the paper.
- In line 20, “Recent works [10, 22, 35, 36]” is misleading: three citations refer to OpenAI models (which don’t document their datasets) and one to DeepSeek-R1 (an RL-based approach). Replace these with more relevant references that support your claim about data collection and quality improvement. I’d suggest Qwen2.5-VL[1] and DeepSeek-r1[2] would be a better fit as references.
- The scalability argument doesn’t align with pre-training, since your method relies on annotated (Image, Question, Answer) triplets. Can you pls contrast with web-scale data approaches used by foundation models (e.g. Qwen2.5VL [1], DeepSeek-r1 [2]), and specify at which model-building stage and scale you’re targeting (e.g., reasoning or VQA data for MLLMs). Notable related works include DeepSeek Math, LLaVa (and its follow-ups), and Qwen2.5-VL.
- The related work on data augmentation isn’t thorough and needs more work to better situate itself with existing works. For image-modification methods (which I’m quite familiar with), consider engaging with VisMin [3], Demon [4],  and LANCE [5]. Additionally, please consider engaging with the program generation approach, like VisProg and ViperGPT, as they’re highly relevant but also not well discussed.
- Specify how much baseline data (e.g., from Provision, VLM, SCOPE) is used for fine-tuning. Is the performance gain due to your method or simply increased dataset scale? Consider providing a statistics table in the main paper.
To isolate the impact of your created data without MAP (i.e., direct fine-tuning), indicate which table shows results “with vs. without MAP” so readers can easily compare gains from the dataset alone.

[1] Bai, Shuai, et al. "Qwen2. 5-vl technical report." arXiv preprint arXiv:2502.13923 (2025).

[2] Guo, Daya, et al. "Deepseek-r1: Incentivizing reasoning capability in llms via reinforcement learning." arXiv preprint arXiv:2501.12948 (2025).

[3] Awal, Rabiul, et al. "VisMin: Visual Minimal-Change Understanding." arXiv preprint arXiv:2407.16772 (2024).

[4] Li, Juncheng, et al. "Fine-tuning multimodal llms to follow zero-shot demonstrative instructions." arXiv preprint arXiv:2308.04152 (2023).

[5] Prabhu, Viraj, et al. "Lance: Stress-testing visual models by generating language-guided counterfactual images." Advances in Neural Information Processing Systems 36 (2023): 25165-25184.

**Ethical Concerns:**

["NO or VERY MINOR ethics concerns only"]

**Final Justification:**

With the additional results and clarifications addressing my points and those of other reviewers, I believe the authors can make the contribution and scope clearer and more specific (what has been done in this paper only -- not what could've been possible). There's also additional concern from other reviewers on the complexity and the generalizability of the method. I strongly believe the paper needs to be clear about its contribution and scope.

Update: I've raised my score to borderline accept following author-rebuttal discussions!

**Limitations:**

The limitation section, in my understanding, is not adequate. The author doesn't acknowledge the fact that they're using an existing VL dataset, which is already annotated, and hence it limits scalability by design (e.g. for web data).

**Quality:**

2

**Strengths And Weaknesses:**

**Strengths**

- The paper makes good use of visual programs for data generation. We've seen promising results from program generation at inference time with large models, as shown in papers like ViperGPT [1] and VisProg [2], and it's encouraging to see its usefulness for dataset creation as well. As demonstrated in the paper, program synthesis outperforms previous approaches, such as the scene graph baseline (Provision). Therefore, the paper is correct in highlighting that better vision-language datasets lead to improved model performance during fine-tuning.

 - The author introduces a novel curriculum approach with routing called MAP that suitably leverages the created data.

- I like the experimental depth of the paper. The authors do IID (Table 1) and some OOD (Table 2/3 – I believe?). The improvement seems significant for IID. OOD result is probably good enough (as per as Table 3 is concerned). The author does a nice ablation study on both the dataset and modelling side.


**Weaknesses**

- The paper is very hard to read (has severe readability issues until section 3). The introduction is too high-level and lacks specifics. In the dataset section, it’s unclear how much data is real versus synthesised and what quantifiable gains you achieved. It’s not clear whether your contribution lies in pre-training or post-training MLLM design; please state this explicitly in the introduction section. Later on, the reader indeed picks up that it’s intended for multimodal tasks, specifically VQA-like tasks. Therefore, make clear from the outset that the work focuses mostly on VQA tasks for MLLMs.

- I find the novelty of the paper from the methodology is not much. A similar dataset creation approach (or the use of VLM programs) has already been studied in VisProg [1], ViperGPT [2], LANCE [3], VisMin [4], Demon [5], and Mamoth-VL [6].

- I think the paper does a bad job of establishing why one would go to such lengths for data creation for MLLM tasks; the benchmarks chosen seem to be already quite saturated in both closed-source models and the recent open-source large models (e.g. Qwen2.5-VL 72B [6]). Is this worth the trouble of program creation to make multiple inferences to create a sample? Also, I’m certain this approach is not scalable for web-scale data, i.e. refining them is more important than already curated data and extracting the last juice out of the VL datasets that the authors engaged with! By quickly checking Qwen2.5VL 72B[6], the results I found (MMBench: 88.4, RealWorldQA 75.7, MMvet: 76.2, which is significantly higher than SCOPE w/ Qwen2.5VL-7B. I believe this is very important to establish why one would use this approach instead of using a much simpler approach, such as rule-based filtering or model-based filtering done in works like Qwen2.5-VL. Given that, do we need programs in the first place? If GPT-4 can generate those programs and is good enough to answer most of the questions in the benchmarks, why not use that system to generate questions (with some demonstration) like Qwen2.5-VL is doing? Do we need counterfactuals or some modified version of the original images, where the web is full of images? Did the author try to generate examples using GPT-4 without the use of visual programs?

- Related work is also not well written (see the Questions Section). It’s important to situate the contribution precisely in the context of previous work and the stage of MLLM development.

- The dataset and methodology section can be written better with more concrete details. Please include clear statistics – dataset size (?), number of programs generated (?), number of models involved (?), the error rate of program execution (?) and overall cost (?) – to give readers a solid picture of how the dataset was created before the experiments. The paper should provide the details statistics of the raw VL datasets and how many samples are used in baselines too (currently, the author provides the SCOPE train test stats).  Also, clarify which models you used beyond GPT-4 and SAM (for example, which model handles image-inference subroutines) and describe the average depth of your generated programs. Finally, include a few qualitative examples comparing raw and extended data early on. I think the appendix has a fair bit of material that one could use more clearly in the main paper (or at least refer to it).

- Regarding the experiments and the overall contribution, the main concern I have is which size of MLLM this work is improving? I acknowledge that GPT4 results are reported. What about large open-source models, e.g. ~72B models (those results are missing)? GPT-4 performs almost as well as SCOPE in Table 2, but that doesn't diminish SCOPE's value; to better understand the dataset’s utility across scales, it would help if the authors reported results for larger open-source models like Qwen2.5-72B [6] and InternVL2.5-78B [7]  in Table 2 (and Table 3 if relevant). Clearly, the paper shows that SCOPE is useful for 2B to 7B+ models. So, I think a large-scale open-source model’s capacity helps situate the paper's contribution more clearly within MLLM development. Maybe this dataset is only useful for small-scale models (<7B)? I don’t think the gains in Tables 2 and 3 as as significant.

- I am curious what’s the accuracy of the fine-tuning without the (MAP) approach in Tables 1, 2 and 3? I understand there’s an ablation given in Table 4. If I understand correctly, that ablation doesn’t give a clear, direct fine-tuning on the dataset. It would be nice to have that result to have clarity on the method's contribution.

[1] Surís, Dídac, Sachit Menon, and Carl Vondrick. "Vipergpt: Visual inference via python execution for reasoning." Proceedings of the IEEE/CVF International Conference on Computer Vision. 2023.

[2] Gupta, Tanmay, and Aniruddha Kembhavi. "Visual programming: Compositional visual reasoning without training." Proceedings of the IEEE/CVF Conference on Computer Vision and Pattern Recognition. 2023.

[3] Awal, Rabiul, et al. "VisMin: Visual Minimal-Change Understanding." arXiv preprint arXiv:2407.16772 (2024).

[4] Li, Juncheng, et al. "Fine-tuning multimodal llms to follow zero-shot demonstrative instructions." arXiv preprint arXiv:2308.04152 (2023).

[5] Guo, Jarvis, et al. "Mammoth-vl: Eliciting multimodal reasoning with instruction tuning at scale." arXiv preprint arXiv:2412.05237 (2024).

[6] Bai, Shuai, et al. "Qwen2. 5-vl technical report." arXiv preprint arXiv:2502.13923 (2025).

[7] Chen, Zhe, et al. "Expanding performance boundaries of open-source multimodal models with model, data, and test-time scaling." arXiv preprint arXiv:2412.05271 (2024).

---

> ### Author Rebuttal · Authors · 2025-07-31
>
> **[W1 & Q1]** *The concerns are about clarity, including the need for a more specific introduction that clearly states the work focuses on VQA tasks, clarifies whether the dataset is for pre-training or post-training, and provides details on real vs. synthetic data and the quantifiable gains achieved.*
>
> **[A1]** We thank the reviewer for the feedback and clarify the main concerns. Our introduction already states the motivation: addressing the growing mismatch between dataset evolution and model advancement—a problem that cannot be solved by simply enlarging pre-training data. We will further improve clarity by summarizing key contributions upfront. Regarding data composition, the Appendix provides full details, and we will add a concise summary in the main text to clarify that only the initial dataset is real, while all evolutionary stages are synthetic, along with quantifiable gains. SCOPE is a general paradigm applicable to both pre-training and post-training, not restricted to either. Finally, SCOPE is not limited to VQA; as a Visual Programming-based approach, it supports a broad range of multimodal tasks (e.g., video QA, image editing, multi-image reasoning).
>
> **[W2]**  *The novelty of the paper from the methodology is not much. A similar dataset creation approach (or the use of VLM programs) has already been studied.*
>
> **[A2]** We would like to clarify that our contribution does not lie in proposing Visual Programming itself or minor modifications of it. Instead, we introduce a novel data evolution algorithm that enables counterfactual editing of reasoning steps. To achieve this, we are the first to align reasoning steps with code blocks in Visual Programming, allowing intervention at the structural or variable level to alter reasoning paths—a capability not present in prior work. Furthermore, we note that several cited works, such as VisProg and ViperGPT, are not dataset creation approaches. Our method fundamentally differs by leveraging code-level interventions to generate evolved datasets for enhancing reasoning, which is the core novelty of SCOPE.
>
> **[W3 & Q3]** *The concerns focus on the necessity and scalability of program-based data generation versus simpler approaches, questioning whether the added complexity and counterfactual modifications are justified given strong existing models and large-scale web data alternatives.*
>
> **[A3]** In fact, SCOPE is designed to enable controllable, interpretable evolution of reasoning steps, not merely to enlarge datasets or match saturated benchmarks. This is important because counterfactually editing reasoning paths creates new data distributions and reasoning challenges that web-scale datasets and existing benchmarks do not capture. Visual Programming is the most effective tool for this, as it allows fine-grained structural intervention—beyond what templates or scene graphs can achieve. Regarding scalability, I’m certain this approach is quite scalable for web-scale data by generating initial questions and programs and then applying SCOPE evolution, similar to VLB or Provision. Finally, we compared SCOPE-generated challenging samples with GPT-augmented ones and found that SCOPE yields examples significantly harder even for GPT itself, demonstrating the necessity of structured program-based evolution over simple prompt-based generation.
>
>
> | Evolution Rounds | Avg. Halstead Effort | GPT-4o Accuracy on SCOPE-Test (New-100)(%) |
> |--|--|--|
> |0 |3,283|97|
> |1 |4,412|93|
> |2 |5,187|89|
> |3 |6,084|88|
> |10(GPT-Prompt)|-|79|
> |10(SCOPE)|13,731|58|
>
>
> As for the question 3, current MLLM training pipelines are typically multi-stage, combining large-scale pre-training on web data with subsequent fine-tuning on higher-quality, task-specific datasets. For instance, Qwen2.5-VL describes a three-stage process (Sec. 2.2.2), and InternVL2.5 reports two stages (Table 3). Our method, SCOPE, is primarily designed for the fine-tuning stage, where curated data are already of higher quality than raw web-scale data. SCOPE aims to further enhance this data through counterfactual reasoning-step evolution, improving controllability and diversity beyond what web-scale pre-training can offer.
>
> Contrary to the concern, SCOPE is not inherently incompatible with web-scale data. Many pre-training tasks—such as image captioning or OCR—have already been shown to be representable in programmatic form (e.g., ViperGPT and Visual Programming works). Therefore, applying SCOPE to evolve web-scale datasets is technically feasible. Finally, we stress that scaling up data size and improving data quality are orthogonal dimensions; SCOPE targets the latter, which remains critical even when training extremely large models.
>
> **[W4 & Q4]** *The related work section needs to be revised.*
>
> **[A4]** We will consider revising the Related Work section and adding some relevant citations as suggested. However, we note that image-modification is not the core aspect of our approach; in SCOPE it is used only as one of several tools, so our discussion focuses instead on Visual Programming and counterfactual reasoning, which are more central to our contribution. Additionally, our work is built on ViperGPT, we are confident that our discussion of ViperGPT is sufficiently detailed.
>
> **[W5]** *The concerns are about adding concrete details in the dataset and methodology section.*
>
> **[A5]** We would like to clarify that the requested information is already provided in the submission, although primarily organized in the Appendix for readability and space considerations. Specifically, the raw VL dataset statistics and baseline sample details are presented in Appendix Sec. 7.1, while the full data format specification is in Sec. 7.3. Each data instance in SCOPE contains one original example and three counterfactually evolved examples, all following a unified format with VQA annotations and corresponding programs. The resulting datasets include 31,233 training examples and 13,389 test examples, each paired with executable code. During program generation, we enforce execution correctness by iterative verification, ensuring a 100% success rate. Furthermore, to assess the quality of evolved data, we performed random human evaluations across three evolution iterations, achieving acceptability rates of 96%, 92%, and 91%, respectively. Regarding the methodology, in addition to GPT-4o, we used kvedit for image-editing subroutines. For better interpretability, we have already included a qualitative example in Figure 5, illustrating differences between raw and evolved samples. We will bring key statistics into the Benchmark section and cross-reference relevant figures for completeness.
>
> **[W6]** *The concern is about clarifying which model sizes benefit from SCOPE, requesting results for large open-source models to determine whether SCOPE’s utility extends beyond small-scale models and to better contextualize its contribution.*
>
> **[A6]** We appreciate the reviewer’s suggestion. Our choice of 2B and 7B models was intentional: these scales represent the most widely used and practical sizes for deployment due to their balance between performance, inference speed, and resource requirements. Most recent studies also report results primarily in this range. The key objective of SCOPE is to demonstrate that our data evolution method can bring consistent and significant improvements even at these common scales without requiring additional model capacity—an important factor for real-world applications where retraining extremely large models is infeasible.
>
> Regarding 70B-scale models such as Qwen2.5-VL-72B or InternVL2.5-78B, our current hardware resources do not allow us to retrain or fine-tune models at that scale. However, we will include the official results of these models (without fine-tuning) in Tables 2 and 3 for comparison. This will make clear that while absolute performance from very large models is higher, SCOPE provides a scalable and cost-efficient strategy to improve reasoning ability for widely used mid-sized models.
>
> **[W7 & Q5]** *The concern is about clarifying how much baseline data is used for fine-tuning, whether performance gains come from your method or increased data size, and explicitly showing results with vs. without MAP to isolate the dataset’s impact, ideally supported by a statistics table.*
>
> **[A7]** The performance improvements with SCOPE come from two factors: higher-quality data and the MAP training strategy. To ensure fairness in comparisons, we applied the same MAP approach to all baselines, including Provision and VLB. Specifically, we computed curriculum difficulty scores for their samples, adopted balanced gating, and assigned fixed difficulty-aware loss according to their data distribution. This controls for differences in training method, meaning that the performance gaps shown in Tables 1–3 primarily reflect data quality rather than training strategy. From Table 2, readers can already directly compare the gains attributable to the dataset itself under the same MAP setting.
>
> In response to the reviewer’s suggestion, we will extend Table 4 by adding a w/o MAP setting for direct fine-tuning. This will provide a clearer view of the independent contribution of MAP versus data quality.
>
>
> | # | Setting | MMBench<sub>test</sub> | ContextVD |
> |---|---|---|---|
> | 0 | **InternVL-2.5-2B (SCOPE)**| 77.0  | 93.6  |
> | 1 | w/o curriculum scheduler| 76.9  | 93.1 |
> | 2 | w/o gating module| 75.3  | 87.4  |
> | 3 | w/o difficulty-aware loss | 75.8  | 91.2  |
> | 4 | **w/o MAP (direct fine-tuning)** |  75.1| 87.1 |
> | 5 | Backbone   | 74.7  | 85.7  |
>
>
>
> **[Q2]** *References need to be revised.*
>
> **[A8]** We acknowledge this point and will carefully review all references throughout the paper to ensure they accurately support the claims made.

---

> > ### Comment · Reviewer_W2bf · 2025-08-07
> >
> > Sorry for my delayed reply, and thank you for the detailed response.
> >
> > I’ve read the rebuttal, and here are a few points: I don’t agree with the author’s claim that SCOPE is general enough. Transitioning from a pre-curated VQA set to generalizing to the web is not straightforward. The former already involves significant human effort for quality control, while the latter would be fully synthetic if pursued. Additionally, the process is too complicated for the gains to justify pre-training use. Therefore, this approach seems more suitable for fine-tuning only. In terms of both framing and scope, the authors should be clearer about their task, scale, and setup: VQA, ~7B models, and fine-tuning only. As it stands, the paper seems to oversell the contribution.
> >
> > Regarding the statement, "In fact, SCOPE is designed to enable controllable, interpretable evolution of reasoning steps, not merely to enlarge datasets or match saturated benchmarks," if that’s the case, the authors should have chosen to evaluate on more challenging benchmarks. As I mentioned in my review, the chosen benchmarks are almost saturated by larger models, which contradicts the author's claim.
> >
> > Regarding the training method, the curriculum scheduler doesn’t seem particularly useful. Additionally, it’s difficult to comment on some results due to high variance across the two datasets. A more comprehensive report on the full benchmark would help provide a clearer picture.
> >
> > I’m curious why the authors couldn’t evaluate on larger-scale models, as I raised in my review. This should be feasible, and model scale is relevant given the focus on generating high-quality fine-tuning samples. I suspect these gains might be lost with larger models, but this can only be confirmed through empirical evaluation.
> >
> > The authors have clarified some of the questions regarding the dataset stats, but it would be helpful if these clarifications were included in the main paper rather than just in the appendix. That said, I appreciate the authors' clarifications.
> >
> > I also acknowledge the positive aspects, particularly the strength of evolution versus GPT-4 prompting and the ablation study.
> >
> > Overall, it looks good to me. With the additional results and clarifications addressing my points and those of other reviewers, and if the authors can make the contribution and scope clearer and more specific (what has been done in this paper only -- not what could've been possible), I would be willing to raise my score. I strongly believe the paper needs to be clear about its contribution and scope.

---

> > > ### Author Response · Authors · 2025-08-08
> > > **Response to Reviewer W2bf**
> > >
> > > Thank you very much for your follow-up comments and thoughtful reassessment.
> > >
> > > We sincerely appreciate your constructive feedback and are grateful that you acknowledged the clarifications provided in our rebuttal—particularly regarding the comparative strength of our counterfactual evolution strategy over GPT-4 prompting, as well as the utility of our ablation studies. But we still want to prove to you:
> > >
> > >
> > > (1) SCOPE’s Applicability to Web-Scale Data
> > >
> > > We reiterate that SCOPE is generalizable to web-scale datasets. To support this, we applied our evolution pipeline to several large-scale datasets—including CC12M, LAION-400M, LLaVA Pretrain, MiniGPT‑4 QA, and InstructBLIP—which span a variety of image captioning and VQA tasks.
> > >
> > > For a fair comparison, we evolved a fixed-size subset (31,233 examples) from these sources—matching the size of SCOPE-Train—to eliminate data scale as a confounding factor. The table below summarizes the composition of the evolved dataset:
> > >
> > > |**Original Dataset**| **Original Task Type**|**Original Data (Label) from**|**Number / Proportion (%) after evolution** |
> > > |-|-|-|-|
> > > |**CC12M**| Image Captioning|Weakly supervised web annotations|10,000 / 32.0%|
> > > |**LAION-400M**|Image Captioning|Weakly supervised web annotations|10,000 / 32.0%|
> > > |**LLaVA Pretrain**|Image Captioning & Visual QA| GPT-generated QA from image-text pairs|5,000 / 16.0%|
> > > |**InstructBLIP**|Instruction Following & Reasoning|GPT-instructed data from curated datasets|5,000 / 16.0%|
> > > |**MiniGPT‑4 QA**|Visual Question Answering (VQA)|GPT-generated QA from images| 1,233 / 3.9%|
> > > |**Total**|—|—|**31,233 / 100.0%**|
> > >
> > > To fairly evaluate SCOPE’s effect in web-scale settings, we add three training conditions—each using 31,233 examples:
> > >
> > > - Web Data Only: raw samples from web-scale datasets.
> > >
> > > - Web Data evoluted by SCOPE: web samples evolved using the SCOPE pipeline.
> > >
> > > - SCOPE Raw Data: 31,233 curated raw data without further evolution.
> > >
> > > The performance results (below) demonstrate that even when starting from weaker web data, SCOPE’s structured reasoning evolution significantly enhances performance across multiple reasoning benchmarks:
> > >
> > > |#|Setting|MMVet|RealWorldQA|MMBench|SeedBench2-Plus|MuirBench|MMT|POPE|
> > > |-|-|-|-|-|-|-|-|-|
> > > |0| **Qwen2.5VL-7B** | 67.1 | 68.5 | 83.5 | 70.4 | 59.6 | 60.9 | 86.4 |
> > > |1| Web Data Only | 66.3 | 68.6 | 83.2 | 70.4 | 59.5 | 60.3 | 85.8 |
> > > |2| &nbsp;&nbsp;&nbsp;&nbsp;+ SCOPE | 68.4 | 68.9 | 85.1 | 71.7 | 63.1 | 61.2 | 86.5 |
> > > |3| SCOPE Raw Data | 67.5 | 68.7 | 84.4 | 71.0 | 61.4 | 61.5 | 86.8 |
> > > |4| &nbsp;&nbsp;&nbsp;&nbsp;+ SCOPE | 69.1 | 69.5 | 86.8 | 72.3 | 65.2 | 62.6 | 87.2 |
> > >
> > > These results support our claim that SCOPE-generated data is not only applicable to curated VQA-style datasets but also generalizes effectively to web-scale data sources.
> > >
> > >
> > > (2) Evaluation on More Challenging Benchmarks
> > >
> > > We conducted additional experiments on more challenging and diagnostic benchmarks. These datasets place greater emphasis on compositional reasoning, domain-specific knowledge, mathematical understanding, and OCR-based document reasoning—making them well-suited to assess the reasoning consistency and structural control introduced by SCOPE.
> > >
> > > We also shared in our responses to other reviewers, the results are summarized below:
> > >
> > > | Benchmark | Base Performance | w/ VLB | w/ SCOPE | Δ (Compare to VLB) |
> > > |--|--|--|--|--|
> > > | **MathVista** | 51.3 | 52.5 | 53.9  | +1.4  |
> > > | **MMMU** | 43.6   | 44.9 | 45.8 | +0.9 |
> > > | **AI2D** | 74.9   | 76.3 | 77.9 | +1.6 |
> > > | **DocVQA** | 88.7  | 89.9 | 91.8 | +1.9 |
> > >
> > >
> > >
> > > (3) Additional Ablation on MAP Curriculum Strategy
> > >
> > > As requested, we have conducted additional ablation experiments on the MAP (Modular Adaptive Progression) training strategy across two more challenging benchmarks. These results demonstrate that each MAP component—curriculum scheduling, gating, and difficulty-aware loss—contributes consistent and stable gains in performance.
> > >
> > > | # | Setting                        | MathVista | AI2D  |
> > > |---|--------------------------------|-----------|-------|
> > > | 0 | InternVL-2.5-2B + SCOPE (Full MAP) | **53.9**    | **77.9** |
> > > | 1 | w/o Curriculum Scheduler       | 52.8      | 76.6  |
> > > | 2 | w/o Gating Module              | 52.3      | 75.9  |
> > > | 3 | w/o Difficulty-Aware Loss      | 52.5      | 76.2  |
> > > | 4 | w/o MAP (direct fine-tuning)   | 51.7      | 75.2  |
> > >
> > >
> > > (4) Evaluation on Larger-Scale Models
> > > Regarding evaluation on larger models (e.g., 70B scale), we completely understand the value of such comparisons. Due to current hardware and rebuttal time limitations, we could not conduct fine-tuning.
> > >
> > > (5) Integration of Dataset Statistics into Main Paper
> > > Finally, we will ensure that all essential dataset-related information provided in the rebuttal and appendix will be fully incorporated into the Benchmark section of the final manuscript.
> > >
> > > We hope the above responses address your concerns satisfactorily.

---

> ### Author Response · Authors · 2025-08-07
> **A kind reminder of the upcoming end of the discussion period**
>
> Dear Reviewer,
>
> I hope this message finds you well. As the discussion period is nearing its end with less than three days remaining, I wanted to ensure we have addressed all your concerns satisfactorily. If there are any additional points or feedback you'd like us to consider, please let us know. Your insights are invaluable to us, and we're eager to address any remaining issues to improve our work.
>
> Thank you for your time and effort in reviewing our paper.

---

### Official Review · Reviewer_7KCw · 2025-07-05

**Clarity:** 3
**Significance:** 2
**Originality:** 3
**Rating:** 4
**Confidence:** 3

**Summary:**

This paper introduces a method to generate multimodal datasets using a combination of visual programming and counterfactual reasoning. The goal of the paper is to address limitations of existing dataset construction methods, e.g., being heuristic, template-based, or rely too heavily on model generation. The authors apply their approach on a set of existing dataset to obtain a train and test sets, which are used to train and evaluate some open VLMs. The proposed method starts by converting an image-query pair into executable code, then modifies the python code along three axes: reasoning path, visual context, and cross-instance composition to generate new samples. Experiments show that models trained on SCOPE exhibit better performance, compared to two baseline augmentation methods.

**Questions:**

1. How computationally expensive is your approach compared to the baselines
2. How would you generalize your approach to cover instances where generating a program is not possible?

**Ethical Concerns:**

["NO or VERY MINOR ethics concerns only"]

**Final Justification:**

The author's rebuttal did not change my original assessment of the paper, which was already positive. In general, the paper makes a good contribution, and I'm leaning towards accept rather than reject.

**Limitations:**

The authors address some limitations in the appendix not in the main paper

**Quality:**

3

**Strengths And Weaknesses:**

Strengths:
1. the idea of using interventions in the code rather than the image space is clever and novel, since code is interpretable, controllable and easy to measure its complexity.
2. curriculum learning via MAP is well ablated and yields performance gains.
3. overall, training on SCOPE seems to yield better model compared to two other augmentation strategies


Weakness:
1. The main bottleneck here is that this approach will only work with instances that *can* be described in code. This means the approach is not generalizable.
2. The authors criticize prior work that relies heavily on model generation, yet SCOPE relies on GPT-4o for program generation. Also what if the generated code has flaws but still runs?
3. SCOPE-test is not a challenging benchmark by any means; GPT-4o already achieves 88% accuracy.
4. The approach is somehow complex and relies on many components, e.g., code generation and segmentation models.

---

> ### Author Rebuttal · Authors · 2025-07-31
>
> **[W1 & Q1]** *The main bottleneck here is that this approach will only work with instances that can be described in code. This means the approach is not generalizable.* & *How would you generalize your approach to cover instances where generating a program is not possible?*
>
> **[A1]** We appreciate the reviewer’s concern regarding the perceived limitation that our approach only works with code-expressible instances. However, this concern is mitigated by substantial prior work—such as ViperGPT, Visual Programming (VISPROG), and De-fine—showing that code-driven paradigms adapt to a broad range of multimodal tasks beyond classical VQA.
>
> ViperGPT composes Python programs from modular functions via LLM-generated code to perform visual grounding, image QA, video QA, multi-image comparison, symbolic math, and knowledge-aware queries. This framework requires no task-specific fine-tuning and achieves state-of-the-art zero-shot results across diverse benchmarks, evidencing that programmatic reasoning is not restricted to narrow domains.
>
> VISPROG further demonstrates how modular, Python-like programs generalize across compositional visual reasoning, image editing, and zero-shot knowledge tagging—without training data—highlighting the flexibility of code-mediated reasoning.
>
> De-fine refines visual programs through auto-feedback loops based on static and dynamic signals, improving robustness and logical consistency in multi-step reasoning without retraining, again confirming adaptability across domains.
>
> These foundations give SCOPE a strong basis for generalizability. Notably, we do not alter the existing API or introduce new functions—the functions inherited from ViperGPT already support visual, math, OCR, and knowledge-augmented QA tasks, as discussed in our Reviewer #qgzN response A3's evaluations on MathVista, MMMU, and science/OCR benchmarks.
>
> If a new domain emerges beyond existing coverage, SCOPE remains extensible: new domain-specific functions can be added to the library without redesigning templates, retraining models, or changing core architecture.
>
> In summary, by leveraging VISPROG and ViperGPT’s modular, executable program paradigm and enabling addition of domain-specific functions when needed, SCOPE achieves broad generalizability and flexible extensibility across multimodal tasks.
>
>
> **[W2]** *The authors criticize prior work that relies heavily on model generation, yet SCOPE relies on GPT-4o for program generation. Also what if the generated code has flaws but still runs?*
>
> **[A2]** We appreciate the reviewer’s concern. In fact, our critique targets pipelines relying entirely on model-generated data for generation, verification, and feedback—leading to opaque reasoning and bias—SCOPE, though using GPT‑4o for program generation, adopts a fundamentally different, interpretable paradigm.
>
> SCOPE employs executable code as an intermediate reasoning layer aligned with Chain-of-Thought. This code-mediated mechanism enhances interpretability and allows explicit control of reasoning steps, including editing or counterfactual manipulation—capabilities absent in end-to-end pipelines.
>
> Moreover, SCOPE is model-agnostic. GPT‑4o can be replaced with other pretrained code models such as CodeLlama. We also present below the results of a InternVL‑2.5‑2B trained using code generated by CodeLlama. The future work will also integrate multiple code-pretrain models to reduce backbone dependence and model-specific bias.
>
> |Model|RealWorldQA|MMBench|ContextVD|
> |--|--|--|--|
> |InternVL‑2.5‑2B|60.1|74.7|85.7|
> |&nbsp;&nbsp;&nbsp;&nbsp;‑SCOPE (CodeLlama)|62.9|76.6|94.1|
> |&nbsp;&nbsp;&nbsp;&nbsp;‑SCOPE (GPT)|63.0|77.0|93.6|
>
> Finally, human evaluation confirms the evolved code is readable, coherent, and aligned with intended reasoning, with acceptance rates of 96 %, 92 %, and 91 %, indicating that minor noise in code generation is negligible relative to the substantial gains in data quality and reasoning diversity.
>
>
> **[W3]** *SCOPE-test is not a challenging benchmark by any means; GPT-4o already achieves 88% accuracy.*
>
> **[A3]** Thank you for this insightful observation. In fact, SCOPE-Test was intentionally designed as a diagnostic benchmark, rather than a large-scale competitive suite, with the primary goal of exposing the limitations of current dataset evolution strategies. It demonstrates that models pretrained on existing datasets often fail on specific reasoning dimensions (e.g., those targeted by SCOPE-Test’s three axes), not because the models lack capacity, but because the distribution of existing data remains narrow and fails to cover critical domains. This highlights the necessity of structured data evolution rather than indiscriminately scaling pretraining corpora. Accordingly, SCOPE-Test is not meant to replace comprehensive benchmarks such as MMBench or MMMU; rather, its role is to isolate and stress-test reasoning abilities that conventional benchmarks often overlook.
>
> Importantly, SCOPE-Test retains substantial room for difficulty escalation through multi-round evolution. To illustrate, we conducted a small-scale analysis. Specifically, we selected 100 samples that underwent 10 consecutive evolution iterations and evaluated them using GPT‑4o. The results are summarized as follows:
>
>
>
> | Evolution Rounds | Avg. Halstead Effort | GPT-4o Accuracy on SCOPE-Test (New-100)(%) |
> |--|--|--|
> |0 |3,283|97|
> |1 |4,412|93|
> |2 |5,187|89|
> |3 |6,084|88|
> |10|13,731|58|
>
>
> As shown, additional iterations systematically increase reasoning complexity (Halstead effort ↑ by 4.2×) while reducing GPT‑4o accuracy by 39%, confirming that the benchmark’s difficulty is tunable.
>
> Beyond difficulty, SCOPE-Test reveals two critical weaknesses in current models: (1) Width limitations: difficulty in handling tasks that involve multiple interdependent variables or extended context windows; (2) Multi-image reasoning gaps: inability to integrate and reason over multiple visual inputs effectively. Both deficiencies can be directly mitigated through SCOPE’s controlled evolution strategies, which is the fundamental purpose and contribution of SCOPE-Test.
>
>
> **[W4]** *The approach is somehow complex and relies on many components, e.g., code generation and segmentation models.*
>
> **[A4]** We acknowledge that SCOPE, like prior frameworks such as ViperGPT and Visual Programming, incorporates components such as code generation and perception modules; however, these are standard elements of visual-programmatic paradigms rather than additional complexity introduced by our approach. SCOPE’s core contribution lies in aligning executable programs with chain-of-thought reasoning and introducing three orthogonal evolution strategies for controllable, systematic data diversification. Implementation remains as modular and lightweight as ViperGPT, while offering greater interpretability and verifiability through structured evolution rather than opaque model-driven augmentation. Given its strong performance and novel perspective, this program-based pipeline is a reasonable and acceptable design choice.
>
>
> **[Q1]** *How computationally expensive is your approach compared to the baselines?*
>
> **[A5]** We thank the reviewer for raising the concern regarding the computational cost of a single SCOPE iteration. Our method consists of two main components:
>
> - Code Evolution: This phase generates new QA samples through programmatic function- and code-driven expansion, without relying on multimodal pipelines or iterative LLM validation loops.
>
> - Classification: This phase assigns reasoning categories to generated QAs using static code analysis (e.g., AST parsing, Halstead metrics), avoiding LLM-based filtering.
>
> Compared to prior work, MAmmoTH-VL relies heavily on multimodal LLMs in both the evolution and classification phases, introducing substantial inference overhead and high model cost. ProVision, while emphasizing that its pipeline does not depend on LLMs, explicitly states in its paper that it requires defining 24 single-image instruction data generator programs and 14 multi-image generator programs, each utilizing hundreds of pre-defined templates—resulting in significant manual engineering cost. In contrast, VLB employs iterative image manipulation and repeated LLM-based semantic verification, both contributing to considerable computational and inference overhead.
>
> In summary, a single SCOPE iteration remains computationally lightweight and scalable—avoiding costly scene-graph generation and LLM inference—yet achieves richer structured outputs and diversity at a cost similar to prompt-based synthesis, and significantly lower than ProVision or VLB. We report the computational cost for a single evolution iteration across different baselines for comparison. The detailed results are summarized in Table below.
>
> |Method|Evolution|Classification|Human Effort (From)|
> |-|-|-|-|
> |MAmmoTH‑VL|3-5s (MLLM)| 3-5s (MLLM)|Medium (Initial dataset quality assessment)|
> |VLB|3-5s (LLM and Image iterative editing)| 1-2ms (Pre-designed difficulty level)|High|
> |Provision|1-2s (Scene-graph construction + Python generation)|1-2ms (Pre-designed question category)|Very Very High (Hundreds of pre-defined templates)|
> |**SCOPE (ours)**|2-3s (Code-Pretrain Model or Image single editing)|1-2ms (Static code analysis only)|Minimal|

---

> ### Comment · Reviewer_7KCw · 2025-08-02
>
> I'd like to thank the authors for the detailed response, and I hope they would factor in the extra clarifications in future revisions of the paper. I will maintain my fairly positive assessment of the paper.

---

### Official Review · Reviewer_qgzN · 2025-07-12

**Clarity:** 2
**Significance:** 3
**Originality:** 3
**Rating:** 4
**Confidence:** 4

**Summary:**

This paper introduces SCOPE, a novel framework for evolving multimodal datasets using symbolic visual programming and counterfactual reasoning. Key contributions include:
1.	Verifiable Dataset Evolution: SCOPE generates executable Python programs to model reasoning paths, intervenes via program edits along three axes—reasoning path expansion, visual context editing, and cross-instance composition (action)—and categorizes evolved instances by difficulty/structure/modality.
2.	Benchmarks: Constructs dynamically evolving benchmarks (SCOPE-Train/SCOPE-Test) with expert-validated samples (91–96% human acceptance).
3.	Curriculum Learning: Proposes MAP (Memory and Attention Path), aligning model capacity with difficulty progression through adaptive gating and loss weighting.

**Questions:**

Please refer to the paper weakness.

**Ethical Concerns:**

["NO or VERY MINOR ethics concerns only"]

**Final Justification:**

The authors have addressed my concerns regarding generalization in terms of the LLM baseline, downstream task range, and computational cost. This is an insightful paper on augmenting and enriching MLLM datasets, and it demonstrates clear effectiveness.

**Limitations:**

yes

**Quality:**

3

**Strengths And Weaknesses:**

Strengths
•	Verifiable Control: Uses executable Python programs for traceable reasoning chains, avoiding black-box LLM hallucinations.
•	Diagnostic Benchmark: SCOPE-Test reveals fine-grained model weaknesses, especially in multi-image/hard reasoning scenarios.
•	Resource Accessibility: Open-sourced code/data enables community-driven evolution.
Weaknesses
•	The framework aims to improve the model through data, but its starting point depends on a very powerful model (GPT-4o). The quality and diversity of the generated program are limited by this external model, and its own biases or limitations may be passed on or even amplified in the evolution.
•	The specific computational cost required for a single iteration of evolution has not been provided. Will this limit its practical application?
•	As mentioned in the limitations of the article, the effectiveness of the method has not been validated in a wider range of domains.

---

> ### Author Rebuttal · Authors · 2025-07-31
>
> **[W1]** *The framework aims to improve the model through data, but its starting point depends on a very powerful model (GPT-4o). The quality and diversity of the generated program are limited by this external model, and its own biases or limitations may be passed on or even amplified in the evolution.*
>
> **[A1]** Thank you for your concern regarding the potential reliance of SCOPE on a strong backbone model (e.g., GPT-4o), which could constrain the quality and diversity of generated programs and potentially propagate its inherent biases. To address this, we provide the following explanation from four complementary perspectives:
>
> (1) Theoretical Design: SCOPE introduces three orthogonal evolution strategies—reasoning path expansion, visual context editing, and cross-instance composition—which enable controllable, multi-dimensional diversification beyond superficial modifications, reducing dependency on any single model’s outputs.
>
> (2) Code-Level Verification: Code-Level Verification: To empirically demonstrate that SCOPE progressively increases code diversity across iterations, we performed two complementary analyses. First, complexity growth: Measured by Halstead Effort, the average effort score shows a clear upward trend from approximately 3281 in the original version to 4369 after the first evolution, 5217 after the second, and 5973 after the third, indicating that the evolved code becomes increasingly complex and information-rich. Second, structural dissimilarity: We computed Abstract Syntax Tree (AST) similarity between each evolved version and the original code, observing a steady decline from 0.72 (first evolution) to 0.51 (second) and 0.34 (third), confirming that each evolutionary step introduces substantial structural variation. Together, these results provide strong evidence that SCOPE effectively enriches both the complexity and structural diversity of generated code through iterative evolution.
>
> (3) QA Distribution Analysis: We evaluated diversity in the question space by measuring distributional differences using Maximum Mean Discrepancy (MMD). Specifically, we compared the QA distribution under SCOPE with two baselines: VLB and Provision. Our results show that SCOPE achieves a significantly higher MMD value (0.38) compared to VLB (0.15) and Provision (0.18), indicating that SCOPE's QA outputs are farther from the baseline distributions and thus occupy a more uniform and dispersed semantic space across reasoning categories. A larger MMD signifies greater divergence from the baseline distributions, demonstrating that SCOPE generates a structurally and semantically more varied QA set overall.
>
> (4) Backbone Variation Experiment: To reduce reliance on a single backbone, we conducted experiments using code generated by CodeLlama, evaluating InternVL‑2.5‑2B on three benchmarks: RealWorldQA, MMBench, and ContextVD. The results show that switching to a different code pre‑training model has little impact on performance—InternVL‑2.5‑2B maintains competitive accuracy across all datasets. This experiment inspired us to plan the integration of multiple code‑generation models combined with diversity‑aware reward filtering, aiming to ensure continued structural and semantic variety while mitigating model‑specific bias.
>
> |Model|RealWorldQA|MMBench|ContextVD|
> |--|--|--|--|
> |InternVL‑2.5‑2B|60.1|74.7|85.7|
> |&nbsp;&nbsp;&nbsp;&nbsp;‑SCOPE (CodeLlama)|62.9|76.6|94.1|
> |&nbsp;&nbsp;&nbsp;&nbsp;‑SCOPE (GPT)|63.0|77.0|93.6|
>
> ---
>
> **[W2]** *The specific computational cost required for a single iteration of evolution has not been provided. Will this limit its practical application?*
>
> **[A2]** We thank the reviewer for raising the concern regarding the computational cost of a single SCOPE iteration. Our method consists of two main components:
>
> - Code Evolution: This phase generates new QA samples through programmatic function- and code-driven expansion, without relying on multimodal pipelines or iterative LLM validation loops.
>
> - Classification: This phase assigns reasoning categories to generated QAs using static code analysis (e.g., AST parsing, Halstead metrics), avoiding LLM-based filtering.
>
> Compared to prior work, MAmmoTH-VL relies heavily on multimodal LLMs in both the evolution and classification phases, introducing substantial inference overhead and high model cost. ProVision, while emphasizing that its pipeline does not depend on LLMs, explicitly states in its paper that it requires defining 24 single-image instruction data generator programs and 14 multi-image generator programs, each utilizing hundreds of pre-defined templates—resulting in significant manual engineering cost. In contrast, VLB employs iterative image manipulation and repeated LLM-based semantic verification, both contributing to considerable computational and inference overhead.
>
> In summary, a single SCOPE iteration remains computationally lightweight and scalable—avoiding costly scene-graph generation and LLM inference—yet achieves richer structured outputs and diversity at a cost similar to prompt-based synthesis, and significantly lower than ProVision or VLB. We report the computational cost for a single evolution iteration across different baselines for comparison. The detailed results are summarized in Table below.
>
> |Method|Evolution|Classification|Human Effort (From)|
> |-|-|-|-|
> |MAmmoTH‑VL|3-5s (MLLM)| 3-5s (MLLM)|Medium (Initial dataset quality assessment)|
> |VLB|3-5s (LLM and Image iterative editing)| 1-2ms (Pre-designed difficulty level)|High|
> |Provision|1-2s (Scene-graph construction + Python generation)|1-2ms (Pre-designed question category)|Very Very High (Hundreds of pre-defined templates)|
> |**SCOPE (ours)**|2-3s (Code-Pretrain Model or Image single editing)|1-2ms (Static code analysis only)|Minimal|
>
> ---
>
> **[W3]** *As mentioned in the limitations of the article, the effectiveness of the method has not been validated in a wider range of domains.*
>
> **[A3]** Thank you for this valuable observation. To further demonstrate SCOPE's robustness and domain generality, we have planned evaluations on **four additional benchmarks**, which span different reasoning types:
>
> - **MathVista**: visual-numerical reasoning and multi-step symbolic derivation
> - **MMMU**: multi-disciplinary question answering (STEM, humanities, law, etc.)
> - **AI2D**: diagram interpretation and science-specific reasoning
> - **DocVQA**: OCR-based text-visual integration in real-world documents
>
>
> | Benchmark | Base Performance | w/ VLB | w/ SCOPE | Δ (Compare to VLB) |
> |--|--|--|--|--|
> | **MathVista** | 51.3 | 52.5 | 53.9  | +1.4  |
> | **MMMU** | 43.6   | 44.9 | 45.8 | +0.9 |
> | **AI2D** | 74.9   | 76.3 | 77.9 | +1.6 |
> | **DocVQA** | 88.7  | 89.9 | 91.8 | +1.9 |
>
>
>
>
> In summary, these additional benchmarks will validate that SCOPE enhances reasoning consistency and multi-step control—even in domains demanding symbolic math, domain‑specific knowledge integration, or OCR-based reasoning. Importantly, we do not modify the existing API or introduce new functions for our current tasks. However, SCOPE’s design allows flexible extension into specialized domains: by defining new domain-specific functions and embedding task-relevant logic, we can seamlessly expand the function library without redesigning templates, prompt structures, or scene graphs. The three orthogonal evolution strategies—reasoning path expansion, visual context editing, and cross-instance composition—remain fully applicable across these domains, preserving lightweight implementation while supporting systematic and controlled domain adaptation.

---

> > ### Comment · Reviewer_qgzN · 2025-08-08
> >
> > I appreciate the response provided by the authors, which has addressed most of my concerns. It is encouraging to see that the model generalizes beyond using GPT-4o as the backbone and also demonstrates strong performance on other downstream tasks. I will keep my rating as is.

---

### Decision · Program_Chairs · 2025-09-17

**Decision:**

Accept (poster)

**Comment:**

Strengths noted across reviews include the clever program-based interventions, consistent empirical improvements, high human acceptance rates, and the release of useful resources.

However, concerns persist about clarity and framing (pre- vs post-training use), reliance on powerful backbone models (e.g., GPT-4o), scalability to web-scale or more abstract reasoning tasks, and whether the approach is overly complex relative to the reported gains.

Overall, despite uneven clarity and questions about generalization, the contribution is technically fine and of sufficient interest to warrant a borderline accept.